# CLASS-WISE GENERALIZATION ERROR: AN INFORMATION-THEORETIC ANALYSIS

## ABSTRACT

Existing generalization theories of supervised learning typically take a holistic approach and provide bounds for the expected generalization over the whole data distribution, which implicitly assumes that the model generalizes similarly for all the classes. In practice, however, there are significant variations in generalization performance among different classes, which cannot be captured by the existing generalization bounds. In this work, we tackle this problem by theoretically studying the class-generalization error, which quantifies the generalization performance of each individual class. We derive a novel information-theoretic bound for class-generalization error using the KL divergence, and we further obtain several tighter bounds using the conditional mutual information (CMI), which are significantly easier to estimate in practice. We empirically validate our proposed bounds in different neural networks and show that they accurately capture the complex class-generalization error behavior. Moreover, we show that the theoretical tools developed in this paper can be applied in several applications beyond this context.

## 1 INTRODUCTION

Despite the considerable progress towards a theoretical foundation for neural networks (He & Tao, 2020), a comprehensive understanding of the generalization behavior of deep learning is still elusive (Zhang et al., 2016; 2021). Over the past decade, several approaches have been proposed to uncover and provide a theoretic understanding of the different facets of generalization (Kawaguchi et al., 2017; He & Tao, 2020; Roberts et al., 2022). In particular, multiple tools have been used to characterize the expected generalization error of neural networks, such as VC dimension (Sontag et al., 1998; Harvey et al., 2017), algorithmic stability (Bousquet & Elisseeff, 2000; Hardt et al., 2016), algorithmic robustness (Xu & Mannor, 2012; Kawaguchi et al., 2022), and information-theoretic measures (Xu & Raginsky, 2017; Steinke & Zakynthinou, 2020; Wang & Mao, 2023). However, relying solely on the analysis of the expected generalization over the entire data distribution may not provide a complete picture. One key limitation of the standard expected generalization error is that it does not provide any insight into the class-specific generalization behavior, as it implicitly assumes that the model generalizes similarly for all the classes.

**Does the model generalize equally for all classes?** To answer this question, we conduct an experiment using deep neural networks, namely ResNet50 (He et al., 2016) on the CIFAR10 dataset (Krizhevsky et al., 2009). We plot the standard generalization error along with the class-generalization errors, i.e., the gap between the test error of the samples from the selected class and the corresponding training error, for three different classes of CIFAR10 in Figure 1 (left). As can be seen, there are significant variants in generalization performance among different classes. For instance, the model overfits the "cats" class, i.e., large generalization error, and generalizes relatively well for the "trucks" class, with a generalization error of the former class consistently 4 times worse than the latter. This suggests that *neural networks do not generalize equally for all classes.* However, reasoning only with respect to the standard generalization error (red curve) cannot capture this behavior.

Motivated by these observations, we conduct an additional experiment by introducing label noise (5%) to the CIFAR10 dataset. Results are presented in Figure 1 (right). Intriguingly, despite the low level of noise, the disparities between the class-wise generalization errors are aggravated, with some classes generalizing up to 8 times worse than others. Further, as shown in this example, different

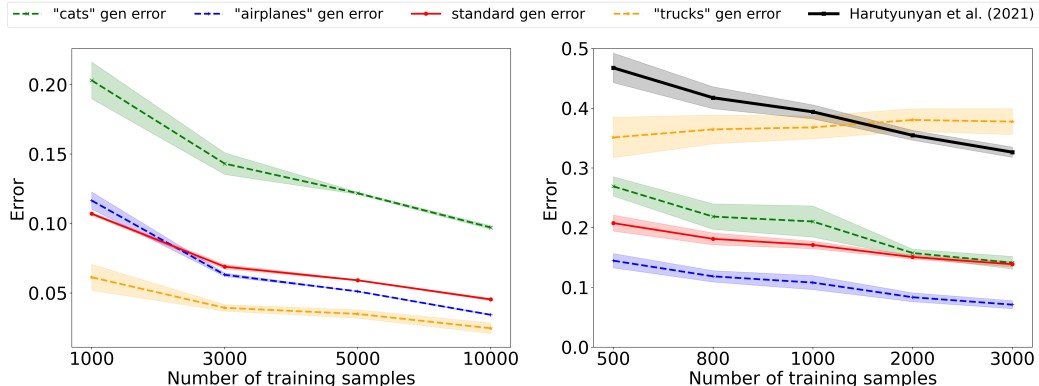

Figure 1: Left: The standard generalization error, i.e., test loss - train loss, and the generalization errors for several classes on CIFAR10 as a function of number of training samples. Right: The standard generalization error, bound proposed by Harutyunyan et al. (2021), and the generalization errors for several classes on noisy CIFAR10. Experimental details are available in Section 3.

classes do not exhibit the same trend when the number of training samples increases. For instance, unlike the other classes, the generalization error of the "trucks" class increases when more training data is available. To further illustrate the issue of standard generalization analysis, we plot the information-theoretic generalization bound proposed in Harutyunyan et al. (2021). Although the bound captures the behavior of the standard generalization error well and can be used to explain the behavior of some classes (e.g., "cat"), it becomes an invalid upper bound for the "trucks" class[1].

When comparing the results on both datasets, it is also worth noting that the generalization error of the same class "trucks" behaves significantly differently on the two datasets. This suggests that class-wise generalization highly depends on factors beyond the class itself, including data distribution, learning algorithm, and the number of training samples. Moreover, in alignment with our findings, Balestriero et al. (2022); Kirichenko et al. (2023) showed that standard data augmentation and regularization techniques, e.g., weight decay and dropout (Goodfellow et al., 2016) improve standard average generalization. However, it is surprising to note that these techniques inadvertently exacerbate the performance variations among different classes.

The main conclusion of all the aforementioned observations is that *neural networks do not generalize equally for all classes, and their class-wise generalization depends on all ingredients of a learning problem*. This paper aims to provide some theoretical understanding of this phenomenon using information-theoretic bounds, as they are both distribution-dependent and algorithm-dependent (Xu & Raginsky, 2017; Neu et al., 2021). This makes them an ideal tool to characterize the class-generalization properties of a learning algorithm. A detailed related work discussion is presented in Appendix A. Our main contributions are as follows:

- We introduce the concept of "class-generalization error," which quantifies the generalization performance of each individual class. We derive a novel information-theoretic bound for this quantity based on KL divergence (Theorem 1). Then, using the super-sample technique by Steinke & Zakynthinou (2020), we derive various tighter bounds that are significantly easier to estimate and do not require access to the model's parameters (Theorems 2, 3, and 4).

- We validate our proposed bounds empirically in different neural networks using CIFAR10 and its noisy variant in Section 3. We show that the proposed bounds can accurately capture the complex behavior of the class-generalization error behavior in different contexts.

- We show that our novel theoretical tools can be applied to the following cases beyond the class-generalization error: (i) study how the class-generalization error affects the standard generalization (Section 4.1); (ii) provide tighter bounds for the subtask problem, where the test data only

---

[1]We note that the bound by Harutyunyan et al. (2021) is proposed for the standard generalization error instead of the class-generalization. Here, we plot it only for illustrative purposes.

encompasses a specific subset of the classes encountered during training (Section 4.2); (iii) derive generalization error bounds for learning in the presence of sensitive attributes (Section 4.3).

**Notations:** We use upper-case letters to denote random variables, e.g., $\mathbf{Z}$, and lower-case letters to denote the realization of random variables. $\mathbb{E}_{\mathbf{Z} \sim P}$ denotes the expectation of $\mathbf{Z}$ over a distribution $P$. Consider a pair of random variables $\mathbf{W}$ and $\mathbf{Z} = (\mathbf{X}, \mathbf{Y})$ with joint distribution $P_{\mathbf{W}, \mathbf{Z}}$. Let $\overline{\mathbf{W}}$ be an independent copy of $\mathbf{W}$, and $\overline{\mathbf{Z}} = (\overline{\mathbf{X}}, \overline{\mathbf{Y}})$ be an independent copy of $\mathbf{Z}$, such that $P_{\overline{\mathbf{W}}, \overline{\mathbf{Z}}} = P_{\mathbf{W}} \otimes P_{\mathbf{Z}}$. For random variables $\mathbf{X}$, $\mathbf{Y}$ and $\mathbf{Z}$, $I(\mathbf{X}; \mathbf{Y}) \triangleq D(P_{\mathbf{X}, \mathbf{Y}} \| P_{\mathbf{X}} \otimes P_{\mathbf{Y}})$ denotes the mutual information (MI), and $I_z(\mathbf{X}; \mathbf{Y}) \triangleq D(P_{\mathbf{X}, \mathbf{Y} | \mathbf{Z}=z} \| P_{\mathbf{X} | \mathbf{Z}=z} \otimes P_{\mathbf{Y} | \mathbf{Z}=z})$ denotes disintegrated conditional mutual information (CMI), and $\mathbb{E}_{\mathbf{Z}}[I_{\mathbf{Z}}(\mathbf{X}; \mathbf{Y})] = I(\mathbf{X}; \mathbf{Y} | \mathbf{Z})$ is the standard CMI. We will also use the notation $\mathbf{X}, \mathbf{Y} | z$ to simplify $\mathbf{X}, \mathbf{Y} | \mathbf{Z} = z$ when it is clear from the context.

## 2 CLASS-GENERALIZATION ERROR

### 2.1 MI-SETTING

Typically, in supervised learning, the training set $\mathbf{S} = \{(\mathbf{X}_i, \mathbf{Y}_i)\}_{i=1}^n = \{\mathbf{Z}_i\}_{i=1}^n$ contains $n$ i.i.d. samples $\mathbf{Z}_i \in \mathcal{Z}$ generated from the distribution $P_{\mathbf{Z}}$. Here, we are interested in the performance of a model with weights $w \in \mathcal{W}$ for data coming from a specific class $y \in \mathcal{Y}$. To this end, we define $\mathbf{S}_y$ as the subset of $\mathbf{S}$ composed of samples only in class $y$. For any model $w \in \mathcal{W}$ and fixed training sets $s$ and $s_y$, the class-wise empirical risk can be defined as follows:

$$L_E(w, s_y) = \frac{1}{n_y} \sum_{(x_i, y) \in s_y} \ell(w, x_i, y), \tag{1}$$

where $n_y$ is the size of $s_y$ ($n_y < n$), and $\ell : \mathcal{W} \times \mathcal{X} \times \mathcal{Y} \to \mathbb{R}_0^+$ is a non-negative loss function. In addition, the class-wise population risk that quantifies how well $w$ performs on the conditional data distribution $P_{\mathbf{X} | \mathbf{Y}=y}$ is defined as

$$L_P(w, P_{\mathbf{X} | \mathbf{Y}=y}) = \mathbb{E}_{P_{\mathbf{X} | \mathbf{Y}=y}}[\ell(w, \mathbf{X}, y)]. \tag{2}$$

A learning algorithm can be characterized by a randomized mapping from the entire training dataset $\mathbf{S}$ to model weights $\mathbf{W}$ according to a conditional distribution $P_{\mathbf{W} | \mathbf{S}}$. The gap between $L_P(w, P_{\mathbf{X} | \mathbf{Y}=y})$ and $L_E(w, s_y)$ measures how well the trained model $\mathbf{W}$ overfits with respect to the data with label $y$, and the expected class-generalization error is formally defined as follows.

**Definition 1.** *(class-generalization error) Given $y \in \mathcal{Y}$, the class-generalization error is*

$$\overline{\text{gen}}_y(P_{\mathbf{X}, \mathbf{Y}}, P_{\mathbf{W} | \mathbf{S}}) \triangleq \mathbb{E}_{P_{\mathbf{W}}}[L_P(\mathbf{W}, P_{\mathbf{X} | \mathbf{Y}=y})] - \mathbb{E}_{P_{\mathbf{W}, \mathbf{S}_y}}[L_E(\mathbf{W}, \mathbf{S}_y)], \tag{3}$$

*where $P_{\mathbf{W}}$ and $P_{\mathbf{W}, \mathbf{S}_y}$ are marginal distributions induced by the learning algorithm $P_{\mathbf{W} | \mathbf{S}}$ and data generating distribution $P_{\mathbf{S}}$.*

**KL divergence bound** For most learning algorithms used in practice, the index of training samples $i$ will not affect the learned model. Thus, we assume that the learning algorithm is symmetric with respect to each training sample. Formally, this assumption can be expressed as follows:

**Assumption 1.** *The marginal distribution $P_{\mathbf{W}, \mathbf{Z}_i}$ obtained from the joint distribution $P_{\mathbf{W}, \mathbf{S}}$, satisfying $P_{\mathbf{W}, \mathbf{Z}_i} = P_{\mathbf{W}, \mathbf{Z}_j}, \forall i \neq j$,*

Assumption 1 states that the learned model $\mathbf{W}$ and each sample $\mathbf{Z}_i$ has the same joint distribution $P_{\mathbf{W}, \mathbf{Z}}$. Under this assumption, the following lemma simplifies the class-generalization error.

**Lemma 1.** *Under Assumption 1, the class-generalization error in definition 1 is given by*

$$\overline{\text{gen}}_y(P_{\mathbf{X}, \mathbf{Y}}, P_{\mathbf{W} | \mathbf{S}}) = \mathbb{E}_{P_{\overline{\mathbf{W}}} \otimes P_{\overline{\mathbf{X}} | y}}[\ell(\overline{\mathbf{W}}, \overline{\mathbf{X}}, y)] - \mathbb{E}_{P_{\mathbf{W}, \mathbf{X} | y}}[\ell(\mathbf{W}, \mathbf{X}, y)]. \tag{4}$$

Lemma 1 shows that, similar to the standard generalization error (Xu & Raginsky, 2017; Bu et al., 2020; Zhou et al., 2022), the class-wise generalization error can be expressed as the difference between the loss evaluated under the joint distribution and the product-of-marginal distribution. The key difference is that both expectations are taken with respect to conditional distributions ($\mathbf{Y} = y$).

The following theorem provides a bound for the class-generalization error in Definition 1.

**Theorem 1.** *For $y \in \mathcal{Y}$, assume Assumption 1 holds and the loss $\ell(\overline{\mathbf{W}}, \overline{\mathbf{X}}, y)$ is $\sigma_y$ sub-gaussian under $P_{\overline{\mathbf{W}}} \otimes P_{\overline{\mathbf{X}}|\overline{\mathbf{Y}}=y}$, then the class-generalization error of class $y$ in Definition 1 can be bounded as:*

$$|\overline{\mathrm{gen}_y}(P_{\mathbf{X},\mathbf{Y}}, P_{\mathbf{W}|\mathbf{S}})| \leq \sqrt{2\sigma_y^2 D(P_{\mathbf{W},\mathbf{X}|y}||P_{\mathbf{W}} \otimes P_{\mathbf{X}|\mathbf{Y}=y})}. \tag{5}$$

The full proof is given in Appendix B.1, which utilizes Lemma 1 and Donsker-Varadhan's variational representation of the KL divergence.

Theorem 1 shows that the class-generalization error can be bounded using a class-dependent conditional KL divergence. Theorem 1 implies that classes with a lower conditional KL divergence between the conditional joint distribution and the product of the marginal can generalize better. To our best knowledge, the bound in Theorem 1 is the first label-dependent bound that can explain the variation of generalization errors among the different classes.

We note that our class-generalization error bound is obtained by considering the generalization gap of each individual sample with label $y$. This approach, as shown in Bu et al. (2020); Zhou et al. (2022); Harutyunyan et al. (2021), yields tighter bounds using the mutual information (MI) between an individual sample and the output of the learning algorithm, compared to the conventional bounds relying on the MI of the total training set and the algorithm's output (Xu & Raginsky, 2017).

## 2.2 CMI-SETTING

One limitation of the proposed bound in Theorem 1 is that it can be vacuous or even intractable to estimate in practice, as the bound involves high dimensional entities, i.e., model weights $\mathbf{W}$ and dataset $\mathbf{S}_y$. The conditional mutual information (CMI) framework, as pioneered by Steinke & Zakynthinou (2020), has been shown in recent studies (Zhou et al., 2022; Wang & Mao, 2023) to offer tighter bounds on generalization error. Remarkably, CMI bounds are always finite even if the weights $\mathbf{W}$ are high dimensional and continuous.

In this section, we extend our analysis using the CMI framework. In particular, we assume that there are $n$ super-samples $\mathbf{Z}_{[2n]} = (\mathbf{Z}_1^\pm, \cdots, \mathbf{Z}_n^\pm) \in \mathcal{Z}^{2n}$ i.i.d generated from $P_{\mathbf{Z}}$. The training data $\mathbf{S} = (\mathbf{Z}_1^{\mathbf{U}_1}, \mathbf{Z}_2^{\mathbf{U}_2}, \cdots, \mathbf{Z}_n^{\mathbf{U}_n})$ are selected from $\mathbf{Z}_{[2n]}$, where $\mathbf{U} = (\mathbf{U}_1, \cdots, \mathbf{U}_n) \in \{-1, 1\}^n$ is the selection vector composed of $n$ independent Rademacher random variables. Intuitively, $\mathbf{U}_i$ selects sample $\mathbf{Z}_i^{\mathbf{U}_i}$ from $\mathbf{Z}_i^\pm$ to be used in training, and the remaining one $\mathbf{Z}_i^{-\mathbf{U}_i}$ is for the test.

For a specific class $y \in \mathcal{Y}$, let $n^y = nP(\mathbf{Y} = y)$, the number of supersamples n scaled with the probability of class $y$. We define the class-generalization error in the CMI setting as follows:

**Definition 2.** *(super-sample-based class-generalization error) For any $y \in \mathcal{Y}$, the class-generalization error is defined as*

$$\overline{\mathrm{gen}_y}(P_{\mathbf{X},\mathbf{Y}}, P_{\mathbf{W}|\mathbf{S}}) \triangleq \mathbb{E}_{\mathbf{Z}_{[2n]}}\Big[\frac{1}{n^y}\sum_{i=1}^{n}\mathbb{E}_{\mathbf{U}_i, \mathbf{W}|\mathbf{Z}_{[2n]}}\big[\mathbb{1}_{\{Y_i^{-U_i}=y\}}\ell(\mathbf{W}, \mathbf{Z}_i^{-\mathbf{U}_i}) - \mathbb{1}_{\{Y_i^{U_i}=y\}}\ell(\mathbf{W}, \mathbf{Z}_i^{\mathbf{U}_i})\big]\Big],$$
$$\tag{6}$$

*where $\mathbb{1}_{\{a=b\}}$ is the indicator function, returning 1 when $a = b$ and zero otherwise.*

Similar to Definition 1, the class-generalization error in Definition 2 measures the expected error gap between the training set and the test set relative to one specific class $y$. Compared to the standard generalization error definition typically used in the super-sample setting (Steinke & Zakynthinou, 2020; Zhou et al., 2022), we highlight two key differences: (i) our class-wise generalization error involves indicator functions to consider only samples belonging to a specific class $y$. (ii) Our generalization error is normalized by $n_{\mathbf{Z}_{[2n]}}^y/2$, which is half of the total samples in $\mathbf{Z}_{[2n]}$ with label $y$. In contrast, the risks are averaged over $n$ samples in the standard super-sample setting.

**Class-CMI bound.** The following theorem provides a bound for the super-sample-based class-generalization error using the disintegrated conditional mutual information between $\mathbf{W}$ and the selection variable $\mathbf{U}_i$ conditioned on super-sample $\mathbf{Z}_{[2n]}$.

**Theorem 2** (class-CMI). *Assume that the loss $\ell(w, x, y) \in [0, 1]$ is bounded, then the class-generalization error for class $y$ in Definition 2 can be bounded as*

$$|\overline{\mathrm{gen}_y}(P_{\mathbf{X},\mathbf{Y}}, P_{\mathbf{W}|\mathbf{S}})| \leq \mathbb{E}_{\mathbf{Z}_{[2n]}}\Big[\frac{1}{n^y}\sum_{i=1}^{n}\sqrt{2\max(\mathbb{1}_{\{\mathbf{Y}_i^-=y\}}, \mathbb{1}_{\{\mathbf{Y}_i^+=y\}})I_{\mathbf{Z}_{[2n]}}(\mathbf{W}; \mathbf{U}_i)}\Big].$$

The proof is based on Donsker-Varadhan's variational representation of the KL divergence and Hoeffding's Lemma (Hoeffding, 1994). The key idea of the proof is specifying the function in Donsker-Varadhan's variational representation to match the definition of class-generalization error. Then, using the fact that for a fixed realization $z_{[2n]}$, $\mathbb{1}_{\{y^{\mathbf{U}_i}=y\}}\ell(\mathbf{W}, z_i^{\mathbf{U}_i}) - \mathbb{1}_{\{y^{-\mathbf{U}_i}=y\}}\ell(\mathbf{W}, z_i^{-\mathbf{U}_i}) = \mathbf{U}_i(\mathbb{1}_{\{y_i^-=y\}}\ell(\mathbf{W}, z_i^-) - \mathbb{1}_{\{y_i^+=y\}}\ell(\mathbf{W}, z_i^+))$ coupled with Hoeffding's Lemma yields the desired result. The full proof is provided in Appendix B.2.

Theorem 2 provides a bound of the class-generalization error with explicit dependency on the model weights $\mathbf{W}$. It implies that the class-generalization error depends on how much information the random selection reveals about the weights when at least one of the two samples of $z_i^\pm$ corresponds to the class of interest $y$. Links between overfitting and memorization in the weights have been established in Zhang et al. (2016); Arpit et al. (2017); Chatterjee (2018). Here, we also see that if the model parameters $\mathbf{W}$ memorize the random selection $\mathbf{U}$, the CMI and the class-generalization error will exhibit high values.

**Class-f-CMI bound.** While the bound in Theorem 2 is always finite as $\mathbf{U}_i$ is binary, evaluating $I_{\mathbf{Z}_{[2n]}}(\mathbf{W}; \mathbf{U}_i)$ in practice can be challenging, especially when dealing with high-dimensional $\mathbf{W}$ as in deep neural networks. One way to overcome this issue is by considering the predictions of the model $f_{\mathbf{W}}(\mathbf{X}_i^\pm)$ instead of the model weights $\mathbf{W}$, as proposed by Harutyunyan et al. (2021). Here, we re-express the loss function $\ell$ based on the prediction $\hat{y} = f_w(x)$ as $\ell(w, x, y) = \ell(\hat{y}, y) = \ell(f_w(x), y)$. Throughout the rest of the paper, we use these two expressions of loss functions interchangeably when it is clear from the context.

In the following theorem, we bound the class-generalization error based on the disintegrated CMI between the model prediction $f_{\mathbf{W}}(\mathbf{X}_i^\pm)$ and the random selection, i.e., $I_{\mathbf{Z}_{[2n]}}(f_{\mathbf{W}}(\mathbf{X}_i^\pm); \mathbf{U}_i)$.

**Theorem 3.** *(class-f-CMI) Assume that the loss $\ell(\hat{y}, y) \in [0, 1]$ is bounded, then the class-generalization error for class $y$ in Definition 2 can be bounded as*

$$|\overline{\mathrm{gen}}_y(P_{\mathbf{X}, \mathbf{Y}}, P_{\mathbf{W}|\mathbf{S}})| \leq \mathbb{E}_{\mathbf{Z}_{[2n]}}\left[\frac{1}{n^y}\sum_{i=1}^n \sqrt{2\max(\mathbb{1}_{\{\mathbf{Y}_i^-=y\}}, \mathbb{1}_{\{\mathbf{Y}_i^+=y\}})I_{\mathbf{Z}_{[2n]}}(f_{\mathbf{W}}(\mathbf{X}_i^\pm); \mathbf{U}_i)}\right].$$

The main benefit of the class-f-CMI bound in Theorem 3, compared to all previously presented bounds, lies in the evaluation of the CMI term involving a typically low-dimensional random variable $f_{\mathbf{W}}(\mathbf{X}_i^\pm)$ and a binary random variable $\mathbf{U}_i$. For example, in the case of binary classification, $f_{\mathbf{W}}(\mathbf{X}_i^\pm)$ will be a pair of two binary variables, which enables us to estimate the class-f-CMI bound efficiently and accurately, as will be shown in Section 3.

**Remark 1.** *In contrast to the bound in Theorem 2, the bound in Theorem 3 does not require access to the model parameters $\mathbf{W}$. It only requires the model output $f(\cdot)$, which makes it suitable even for non-parametric approaches and black-box algorithms.*

**Remark 2.** *Both our bounds in Theorems 2 and 3 involve the term $\max(\mathbb{1}_{\{\mathbf{Y}_i^-=y\}}, \mathbb{1}_{\{\mathbf{Y}_i^+=y\}})$, which filters out the CMI terms where neither sample $Z_i^+$ nor $Z_i^-$ corresponds to the class $y$. However, this term does not require both samples $Z_i^\pm$ to belong to class $y$. In the case that one sample in the pair $(Z_i^-, Z_i^+)$ is from class $y$ and the other is from a different class, this term is non-zero and the information from both samples of the pair contributes to the bound ($I_{\mathbf{Z}_{[2n]}}(f_{\mathbf{W}}(\mathbf{X}_i^\pm); \mathbf{U}_i)$). From this perspective, samples from other classes ($\neq y$) can still affect these bounds, leading to loose bounds on the generalization error of class $y$.*

**Class-$\Delta_y L$-CMI bound.** The term $\max(\mathbb{1}_{\{\mathbf{Y}_i^-=y\}}, \mathbb{1}_{\{\mathbf{Y}_i^+=y\}})$ is a proof artifact and makes the bound inconveniently loose. One solution to overcome this is to consider a new random variable $\Delta_y \mathbf{L}_i$ based on the indicator function and the loss, i.e., $\Delta_y \mathbf{L}_i \triangleq \mathbb{1}_{\{y_i^-=y\}}\ell(f_{\mathbf{W}}(\mathbf{X}_i)^-, y_i^-) - \mathbb{1}_{\{y_i^+=y\}}\ell(f_{\mathbf{W}}(\mathbf{X}_i)^+, y_i^+)$. As shown in Wang & Mao (2023); Hellström & Durisi (2022), using the difference of the loss functions on $\mathbf{Z}_i^\pm$ instead of the model output yields tighter generalization bounds for the standard generalization error. We note that $\Delta_y \mathbf{L}_i$ is fundamentally different than $\mathbf{L}_i$ introduced in Wang & Mao (2023). $\mathbf{L}_i$ simply represents the difference in loss functions, while $\Delta_y \mathbf{L}_i$ can be interpreted as a weighted sum of class-dependent losses. The following Theorem provides a bound based on the CMI between this newly introduced variable and the random selection.

**Theorem 4.** *(class-$\Delta_y L$-CMI) Assume that the loss $\ell(\hat{y}, y) \in [0, 1]$, then the class-generalization error of class $y$ defined in 2 can be bounded as*

$$|\overline{\mathrm{gen}_y}(P_{\mathbf{X},\mathbf{Y}}, P_{\mathbf{W}|\mathbf{S}})| \leq \mathbb{E}_{\mathbf{Z}_{[2n]}}\Big[\frac{1}{n^y}\sum_{i=1}^{n}\sqrt{2I_{\mathbf{Z}_{[2n]}}(\Delta_y \mathbf{L}_i; \mathbf{U}_i)}\Big]. \tag{7}$$

*Moreover, we have the $\Delta_y L$-CMI bound is always tighter than the class-$f$-CMI bound in Theorem 3, and the latter is always tighter than the class-CMI bound in Theorem 2.*

Unlike the bound in Theorem 3, the bound in Theorem 4 does not directly rely on the model output $f(\cdot)$. Instead, it only requires the loss values for both $\mathbf{Z}_i^{\pm}$ to compute $\Delta_y \mathbf{L}_i$.

Intuitively, the difference between two weighted loss values, $\Delta_y \mathbf{L}_i$, reveals much less information about the selection process $\mathbf{U}_i$ compared to the pair $f_{\mathbf{W}}(\mathbf{X}_i^{\pm})$. Another key advantage of the bound in Theorem 4 compared to Theorem 3 is that computing the CMI term $I_{\mathbf{Z}_{[2n]}}(\Delta_y \mathbf{L}_i; \mathbf{U}_i)$ is simpler, given that $\Delta_y \mathbf{L}_i$ is a one-dimensional scalar, as opposed to the two-dimensional $f_{\mathbf{W}}(\mathbf{X}_i^{\pm})$.

The result of Theorem 4 provides an interesting interpretation of the class-generalization error in a communication setting. In particular, for fixed $z_{[2n]}$, $P_{\Delta_y \mathbf{L}_i | \mathbf{U}_i}$ specifies a memoryless channel with input $\mathbf{U}_i$ and output $\Delta_y \mathbf{L}_i$. Then, the $\Delta_y L$-CMI term can be interpreted as "the rate of reliable communication" (Shannon, 1948; Gallager, 1968) achievable with the Rademacher random variable $\mathbf{U}_i$ over this channel $P_{\Delta_y \mathbf{L}_i | \mathbf{U}_i}$. From this perspective, Theorem 4 suggests that classes with *lower* rates of reliable communication generalize better.

## 3 EMPIRICAL EVALUATIONS

In this section, we conduct empirical experiments to evaluate the effectiveness of our class-wise generalization error bounds. As mentioned earlier, The bounds in Section 2.2 are significantly easy to estimate in practical scenarios. Here, we evaluate the error bounds in Theorems 3 and 4 for deep neural networks.

We follow the same experimental settings in Harutyunyan et al. (2021), i.e., we fine-tune a ResNet-50 (He et al., 2016) on the CIFAR10 dataset (Krizhevsky et al., 2009) (Pretrained on ImageNet (Deng et al., 2009)). Moreover, to understand how well our bounds perform in a more challenging situation and to further highlight their effectiveness, we conduct an additional experiment with a noisy variant (5% label noise) of CIFAR10. The experimental details are provided in Appendix C.1.

The class-wise generalization error of two classes from CIFAR10 "trucks" and "cats", along with the bounds in Theorems 3 and 4 are presented in the first two columns of Figure 2. The results on all the 10 classes for both datasets are presented in Appendix C.3.

Figure 2 shows that both bounds can capture the behavior of the class-generalization error. As expected, the class-$\Delta_y L$-CMI is consistently tighter and more stable compared to the class-$f$-CMI bound for all the different scenarios. For CIFAR10 in Figure 2 (top), as we increase the number of training samples, the "trucks" class has a relatively constant class-generalization error, while the "cats" class has a large slope at the start and then a steady incremental decrease. For both classes, the class-$\Delta_y L$-CMI precisely predicts the behavior of class-generalization error.

The results on noisy CIFAR10 in Figure 2 (bottom) and the results in Appendix C.3 are consistent with these observations. Notably, the "trucks" generalization error decreases for CIFAR10 and increases for noisy CIFAR10 with respect to the number of samples. Moreover, the class-generalization error of "cat" is worse than "trucks" in CIFAR10, but the opposite is true for the noisy CIFAR10. All these complex behaviors of class-generalization errors are successfully captured by the class-$\Delta_y L$-CMI bound.

As can be seen from the left and middle plots in Figure 2, the class-$\Delta_y L$-CMI bound scales proportionally to the true class-generalization error, i.e., higher class-$\Delta_y L$-CMI bound value corresponds to higher class-generalization error. To further highlight this dependency, we plot in Figure 2 (right) the scatter plot between the different class-generalization errors and their corresponding class-$\Delta_y L$-CMI bound values for the different classes in CIFAR10 (top) and Noisy CIFAR10 (bottom) under different number of samples. We observe that our bound is linearly correlated with the class-generalization error and can efficiently predict its behavior. We also note that $f$-CMI bound exhibits

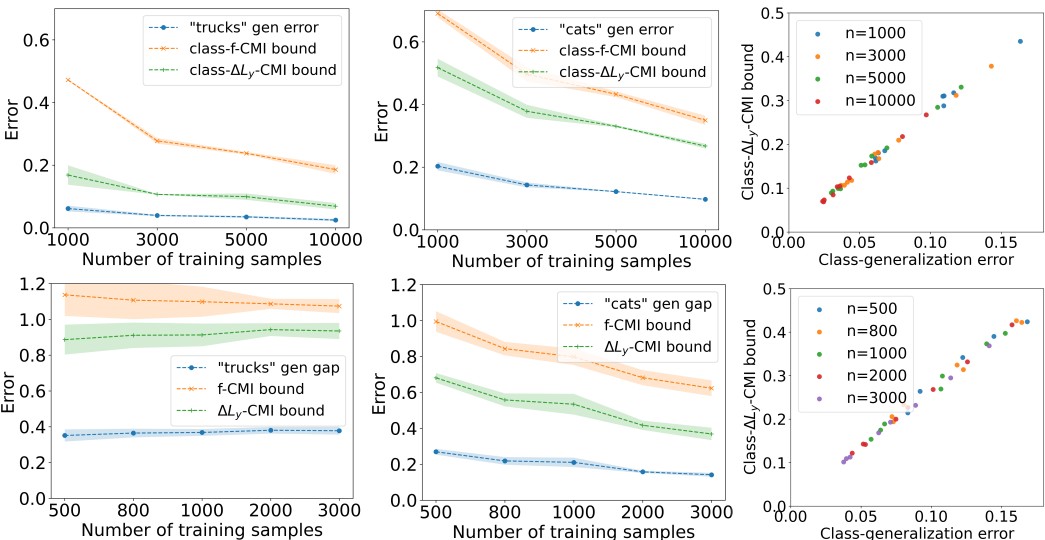

Figure 2: Experimental results of class-generalization error and our bounds in Theorems 3 and 4 for the class of "trucks" (left) and "cats" (middle) in CIFAR10 (top) and noisy CIFAR10 (bottom), as we increase the number of training samples. In the right column, we provide the scatter plots between the bound in Theorem 4 and the class-generalization error of the different classes for CIFAR10 (top) and noisy CIFAR10 (bottom).

a similar behavior as shown in Appendix C.3. This suggests that one can use our bound to predict which classes will generalize better than others. Additional results are provided in Appendix C.5.

## 4 OTHER APPLICATIONS

Besides enabling us to study class-wise generalization errors, the tools developed in this work can also be used to provide theoretical insights into several other applications. In this Section, we explore several use cases for the developed tools.

### 4.1 FROM CLASS-GENERALIZATION ERROR TO STANDARD GENERALIZATION ERROR

In this subsection, we study the connection between the standard expected generalization error and the class-generalization error. We extend the bounds presented in Section 2 into class-dependent expected generalization error bounds.

First, we notice that taking the expectation over $P_{\mathbf{Y}}$ for the class-generalization error defined in 1 yields the standard expected generalization error. Thus, we obtain the first class-dependent bound for the standard generalization error by taking the expectation of $y \sim P_{\mathbf{Y}}$ in Theorem 1.

**Corollary 1.** *Assume that for every $y \in \mathcal{Y}$, the loss $\ell(\overline{\mathbf{W}}, \overline{\mathbf{X}}, y)$ is $\sigma_y$ sub-gaussian under $P_{\overline{\mathbf{W}}} \otimes P_{\overline{\mathbf{X}}|\overline{\mathbf{Y}}=y}$, then*

$$|\overline{\text{gen}}(P_{\mathbf{X},\mathbf{Y}}, P_{\mathbf{W}|\mathbf{S}})| \leq \mathbb{E}_{Y'}\left[\sqrt{2\sigma_{\mathbf{Y}'}^2 D(P_{\mathbf{W}|\mathbf{Z}} \otimes P_{\mathbf{X}|\mathbf{Y}=Y'} || P_{\mathbf{W}} \otimes P_{\mathbf{X}|\mathbf{Y}=Y'})}\right]. \tag{8}$$

We note that in the case where sub-gaussian parameter $\sigma_y = \sigma$ is independent of $y$, we can further show that the bound in 1 is tighter than the individual sample bound in Bu et al. (2020). The proof is available in Appendix C.4. We also extend the results in Theorems 2, 3, and 4 into standard generalization bounds. For example, in Corollary 2, we provide such an extension of Theorem 4.

**Corollary 2.** *Assume that the loss $\ell(\hat{y}, y) \in [0, 1]$, then*

$$|\overline{\text{gen}}(P_{\mathbf{X},\mathbf{Y}}, P_{\mathbf{W}|\mathbf{S}})| \leq \mathbb{E}_{\mathbf{Y}}\left[\mathbb{E}_{\mathbf{Z}_{[2n]}}\left[\frac{1}{n^{\mathbf{Y}}}\sum_{i=1}^{n}\sqrt{2I_{\mathbf{Z}_{[2n]}}(\Delta_{\mathbf{Y}}\mathbf{L}_i; \mathbf{U}_i)}\right]\right]. \tag{9}$$

To the best of our knowledge, Corollaries 1 and 2 are the first generalization bounds to provide explicit dependency on the labels. While prior bounds (Steinke & Zakynthinou, 2020; Harutyunyan et al., 2021; Wang & Mao, 2022; 2023) are tight and efficient to estimate, they fail to provide any insights on how different classes affect the standard generalization error. The results presented here address this gap and provide explicit label-dependent bounds. It shows that the generalization error of an algorithm scales as the expectation of the bound depending only on a single class. Moreover, in classification with $m$ classes, the bounds become a sum of each class-generalization error weighted by the probability of the class, i.e., $P(\mathbf{Y} = y)$. This shows that classes with a higher occurrence probability affect the standard generalization error more. From this perspective, our results can also provide insights into developing algorithms with better generalization capabilities by focusing on the class-generalization error. For example, one can employ augmentation techniques targeted at the classes with higher class-generalization error bounds to attenuate their respective class error and thus improve the standard generalization of the model.

## 4.2 SUB-TASK PROBLEM

Here, "*subtask problem*" refers to a specific case of distribution shift in supervised learning, where the training data generated from the source domain $P_{\mathbf{X},\mathbf{Y}}$ consists of multiple classes, while the test data for the target domain $Q_{\mathbf{X},\mathbf{Y}}$ only encompasses a specific known subset of the classes encountered during training. This problem is motivated by the situation where a large model has been trained on numerous classes, potentially over thousands, but is being utilized in a target environment where only a few classes, observed during training, exist. By tackling the problem as a standard domain adaptation task, the generalization error of the subtask problem can be bounded as follows:

$$\overline{\text{gen}}_{Q,E_P} \triangleq \mathbb{E}_{P_{\mathbf{W},\mathbf{S}}}[L_Q(\mathbf{W}) - L_E(\mathbf{W},\mathbf{S})] \leq \sqrt{2\sigma^2 D(Q_{\mathbf{X},\mathbf{Y}}\|P_{\mathbf{X},\mathbf{Y}})} + \sqrt{2\sigma^2 I(\mathbf{W};\mathbf{S})}, \quad (10)$$

where $L_Q(w) = L_P(w, Q_{\mathbf{X},\mathbf{Y}})$ denotes the population risk of $w$ under distribution $Q_{\mathbf{X},\mathbf{Y}}$. The full details of deriving such bound are available in appendix D.2. We note that (Wu et al., 2020) further tightens the result in equation 10, but these bounds are all based on the KL divergence $D(Q_{\mathbf{X},\mathbf{Y}}\|P_{\mathbf{X},\mathbf{Y}})$ for any generic distribution shift problem and do not leverage the fact that the target task is encapsulated in the source task.

Obtaining tighter generalization error bounds for the subtask problem is straightforward using our class-wise generalization bounds. In fact, the generalization error bound of the subtask can be obtained by summing the class-wise generalization over the space of the subtask classes $\mathcal{A}$. Formally, by taking the expectation of $\mathbf{Y} \sim Q_{\mathbf{Y}}$, we obtain the following notion of the subtask generalization error:

$$\overline{\text{gen}}_{Q,E_Q} \triangleq \mathbb{E}_{Q_{\mathbf{Y}}}\left[\overline{\text{gen}}_{\mathbf{Y}}\right] = \mathbb{E}_{P_{\mathbf{W},\mathbf{S}}}[L_Q(w) - L_{E_Q}(\mathbf{W},\mathbf{S})], \quad (11)$$

where $L_{E_Q}(w, S) = \frac{1}{n_{\mathcal{A}}} \sum_{y_i \in \mathcal{A}} \ell(w, x_i, y_i)$ is the empirical risk relative to the target domain $Q$, and $n_{\mathcal{A}}$ is the number of samples in $S$ such that their labels $y_i \in \mathcal{A}$. We are interested in deriving generalization bounds for $\overline{\text{gen}}_{Q,E_Q}$, as it only differs from $\overline{\text{gen}}_{Q,E_P}$ by the difference in the empirical risk $L_{E_Q}(\mathbf{W},\mathbf{S}) - L_E(\mathbf{W},\mathbf{S})$, which can be computed easily in practice.

Using Jensen's inequality, we have $|\overline{\text{gen}}_{Q,E_Q}| = |\mathbb{E}_{\mathbf{Y}\sim Q_{\mathbf{Y}}}\left[\overline{\text{gen}}_{\mathbf{Y}}\right]| \leq \mathbb{E}_{\mathbf{Y}\sim Q_{\mathbf{Y}}}\left[|\overline{\text{gen}}_{\mathbf{Y}}|\right]$. Thus, we can use the results from Section 2 to obtain tighter bounds. For example, using Theorem 4, we can obtain the corresponding subtask generalization error bound in Theorem 6.

**Theorem 5.** *(subtask-CMI) Assume that the loss $\ell(w,x,y) \in [0,1]$ is bounded, then the subtask generalization error defined in 11 can be bounded as*

$$|\overline{\text{gen}}_{Q,E_Q}| \leq \mathbb{E}_{\mathbf{Y}\sim Q_{\mathbf{Y}}}\left[\mathbb{E}_{\mathbf{Z}_{[2n]}}\left[\frac{1}{n^{\mathbf{Y}}}\sum_{i=1}^{n}\sqrt{2\max(\mathbb{1}_{\mathbf{Y}_i^- = \mathbf{Y}}, \mathbb{1}_{\mathbf{Y}_i^+ = \mathbf{Y}})I_{\mathbf{Z}_{[2n]}}(\mathbf{W};\mathbf{U}_i)}\right]\right].$$

**Theorem 6.** *(subtask-$\Delta L_y$-CMI) Assume that the loss $\ell(w,x,y) \in [0,1]$ is bounded, Then the subtask generalization error defined in 11 can be bounded as*

$$|\overline{\text{gen}}_{Q,E_Q}| \leq \mathbb{E}_{\mathbf{Y}\sim Q_{\mathbf{Y}}}\left[\mathbb{E}_{\mathbf{Z}_{[2n]}}\left[\frac{1}{n^{\mathbf{Y}}}\sum_{i=1}^{n}\sqrt{2I_{\mathbf{Z}_{[2n]}}(\Delta_{\mathbf{Y}}\mathbf{L}_i;\mathbf{U}_i)}\right]\right].$$

Similarly, we can extend Theorems 3 and 2 to use the model's output or weights instead of $\Delta_y\mathbf{L}_i$.

**Remark 3.** *Existing distribution shift bounds, e.g., the bound in equation 10, typically depend on some measure that quantifies the discrepancy between the target and domain distributions, e.g., KL divergence. However, as can be seen in Theorem 6, the proposed bounds here are discrepancy-independent and involve only a quantity analog to the mutual information term in equation 10.*

### 4.3 GENERALIZATION CERTIFICATES WITH SENSITIVE ATTRIBUTES

One main concern hindering the use of machine learning models in high-stakes applications is the potential biases on sensitive attributes such as gender and skin color (Mehrabi et al., 2021; Barocas et al., 2017). Thus, it is critical not only to reduce sensitivity to such attributes but also to be able to provide guarantees on the fairness of the models (Holstein et al., 2019; Rajkomar et al., 2018). One aspect of fairness is that the machine learning model should generalize equally well for each minor group with different sensitive attributes (Barocas et al., 2017; Williamson & Menon, 2019).

By tweaking the definition of our class-generalization error, we show that the theoretical tools developed in this paper can be used to obtain bounds for attribute-generalization errors. Suppose we have a random variable $\mathbf{T} \in \mathcal{T}$ representing a sensitive feature. One might be interested in studying the generalization of the model for the sub-population with the attribute $\mathbf{T} = t$. Inspired by our class-generalization, we define the attribute-generalization error as follows:

**Definition 3.** *(attribute-generalization error) Given $t \in \mathcal{T}$, the attribute-generalization error is defined as follows:*

$$\overline{\text{gen}_t}(P_{\mathbf{X},\mathbf{Y}}, P_{\mathbf{W}|\mathbf{S}}) = \mathbb{E}_{P_{\mathbf{W}} \otimes P_{\mathbf{Z}|\mathbf{T}=t}}[\ell(\overline{\mathbf{W}}, \overline{\mathbf{Z}})] - \mathbb{E}_{P_{\mathbf{W}|\mathbf{Z}} \otimes P_{\mathbf{Z}|\mathbf{T}=t}}[\ell(\mathbf{W}, \mathbf{Z})]. \tag{12}$$

By exchanging $\mathbf{X}$ and $\mathbf{Y}$ with $\mathbf{Z}$ and $\mathbf{T}$ in Theorem 1, respectively, we can show the following bound for the attribute-generalization error.

**Theorem 7.** *Given $t \in \mathcal{T}$, assume that the loss $\ell(\mathbf{W}, \mathbf{Z})$ is $\sigma$ sub-gaussian under $P_{\overline{\mathbf{W}}} \otimes P_{\overline{\mathbf{Z}}}$, then the attribute-generalization error of the sub-population $\mathbf{T} = t$, can be bounded as follows:*

$$|\overline{\text{gen}_t}(P_{\mathbf{X},\mathbf{Y}}, P_{\mathbf{W}|\mathbf{S}})| \leq \sqrt{2\sigma^2 D(P_{\mathbf{W}|\mathbf{Z}} \otimes P_{\mathbf{Z}|\mathbf{T}=t} || P_{\mathbf{W}} \otimes P_{\mathbf{Z}|\mathbf{T}=t})}. \tag{13}$$

We note extending our results to the CMI settings is also straightforward. Using the attribute generalization, we can show that the standard expected generalization error can be bounded as follows:

**Corollary 3.** *Assume that the loss $\ell(\mathbf{W}, \mathbf{Z})$ is $\sigma$ sub-gaussian under $P_{\overline{\mathbf{W}}} \otimes P_{\overline{\mathbf{Z}}}$, then*

$$|\overline{\text{gen}}(P_{\mathbf{X},\mathbf{Y}}, P_{\mathbf{W}|\mathbf{S}})| \leq \mathbb{E}_{\mathbf{T}'}\left[\sqrt{2\sigma^2 D(P_{\mathbf{W}|\mathbf{Z}} \otimes P_{\mathbf{Z}|\mathbf{T}=\mathbf{T}'} || P_{\mathbf{W}} \otimes P_{\mathbf{Z}|\mathbf{T}=\mathbf{T}'})}\right]. \tag{14}$$

The result of Corollary 3 shows that the average generalization error is upper-bounded by the expectation over the attribute-wise generalization. This shows that it is possible to improve the overall generalization by reducing the generalization of each population relative to the sensitive attribute.

## 5 CONCLUSION & FUTURE WORK

This paper studied the deep learning generalization puzzle of the noticeable heterogeneity of overfitting among different classes by introducing and exploring the concept of "class-generalization error". To our knowledge, we provided the first rigorous generalization bounds for this concept in the MI and CMI settings. We also empirically strengthened the findings with supporting experiments validating the efficiency of the proposed bounds. Furthermore, we demonstrated the versatility of our theoretical tools in providing tight bounds for various contexts.

Overall, our goal is to understand generalization in deep learning through the lens of information theory, which motivates future work on how to prevent high class-generalization error variability and ensure 'equal' generalization among the different classes. Other possible future research endeavors focus on obtaining tighter bounds for the class-generalization error and studying this concept in different contexts beyond supervised learning, e.g., transfer and self-supervised learning.

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

## A APPENDIX: RELATED WORK

**Information-theoretic generalization error bounds:** Information-theoretic bounds have attracted a lot of attention recently to characterize the generalization of learning algorithms (Neu et al., 2021; Wang et al., 2010; Aminian et al., 2021; Wu et al., 2020; Wang & Mao, 2022; Modak et al., 2021; Wang & Mao, 2021; Shui et al., 2020; Wang et al., 2023; Alabdulmohsin, 2020). In the supervised learning context, several standard generalization error bounds have been proposed to based on different tools, e.g., KL divergence (Zhou et al., 2023), Wasserstein distance (Rodríguez Gálvez et al., 2021), mutual information between the samples and the weights (Xu & Raginsky, 2017; Bu et al., 2020). Recently, it was shown that tighter generalization bounds could be obtained based on the conditional mutual information (CMI) setting (Steinke & Zakynthinou, 2020; Zhou et al., 2022). Based on this framework, Harutyunyan et al. (2021) derived $f$-CMI bounds based on the model output. In Hellström & Durisi (2022), tighter bounds have been obtained based on the CMI of the loss function, which is further tightened by Wang & Mao (2023) using the $\delta L$ CMI.

**Class-dependent analysis:** Incorporating label information in generalization analysis is not entirely new (He & Su, 2020; Chen et al., 2020; Deng et al., 2021). For example, in He & Su (2020), the questions "When and how does the update of weights of neural networks using induced gradient at an example impact the prediction at another example?" have been extensively studied, and it was observed that the impact is significant if the two samples are from the same class. In Balestriero et al. (2022); Kirichenko et al. (2023); Bitterwolf et al. (2022); Lee et al. (2023), it has been showed that while standard data augmentation techniques (Goodfellow et al., 2016) help improve overall performance, it yields lower performance on minority classes. From a theoretical perspective, (Deng et al., 2021) noticed that in uniform stability context, the sensitivity of neural networks depends highly on the label information and thus proposed the concept of "*Locally Elastic Stability*" to derive tighter algorithmic stability generalization bounds. In Tishby et al. (2000), an information bottleneck principle is proposed which states that an optimal feature map simultaneously minimizes its mutual information with the feature distribution and maximizes its mutual information with the label distribution, thus incorporating the class information. In Tishby & Zaslavsky (2015); Saxe et al. (2019); Kawaguchi et al. (2023), this principle was used to explain the generalization of neural networks.

## B APPENDIX: PROOFS OF THE THEOREMS IN SECTION 2

This appendix includes the missing proofs of the results presented in the main text in Section 2.

**Lemma 2.** *Let $\mathbf{X}$ be a bounded random variable, i.e., $\mathbf{X} \in [a, b]$ almost surely. If $\mathbb{E}[\mathbf{X}] = 0$, then $\mathbf{X}$ is $(b - a)$-subgaussian and we have:*

$$\mathbb{E}[e^{\lambda \mathbf{X}}] \leq e^{\frac{\lambda^2 (b-a)^2}{8}}, \ \ \forall \lambda \in \mathbb{R}. \tag{15}$$

Next, we need the following Lemma 3 as a main tool to prove Theorem 2, 3, and 8 in Section 2.2 in the CMI setting.

**Lemma 3.** *Consider the CMI setting. Let $\mathbf{V} \in \mathcal{V}$ be a random variable, possibly depending on $\mathbf{W}$. For any function g that can be written as $g(\mathbf{V}, \mathbf{U}_i, z_{[2n]}) = \mathbb{1}_{\{y^{\mathbf{U}_i} = y\}} h(\mathbf{V}, z_i^{\mathbf{U}_i}) - \mathbb{1}_{\{y^{-\mathbf{U}_i} = y\}} h(\mathbf{V}, z_i^{-\mathbf{U}_i})$ such that $h \in [0, 1]$ is a bounded function, we have*

$$\mathbb{E}_{\mathbf{V}; \mathbf{U}_i | \mathbf{Z}_{[2n]} = z_{[2n]}}[g(\mathbf{V}, \mathbf{U}_i, z_{[2n]})] \leq \sqrt{2 \max(\mathbb{1}_{\{y_i^- = y\}}, \mathbb{1}_{\{y_i^+ = y\}}) I_{z_{[2n]}}(\mathbf{V}; \mathbf{U}_i)}. \tag{16}$$

*Proof.* Let $(\overline{\mathbf{V}}, \overline{\mathbf{U}_i})$ be an independent copy of $(\mathbf{V}, \mathbf{U}_i)$. The disintegrated mutual information $I_{z_{[2n]}}(\mathbf{V}; \mathbf{U}_i)$ is equal to:

$$I_{z_{[2n]}}(\mathbf{U}_i; \mathbf{V}) = D\big(P_{\mathbf{V}, \mathbf{U}_i | \mathbf{Z}_{[2n]} = z_{[2n]}} \| P_{\mathbf{V} | \mathbf{Z}_{[2n]} = z_{[2n]}} P_{\mathbf{U}_i}\big), \tag{17}$$

Thus, by the Donsker–Varadhan variational representation of KL divergence, $\forall \lambda \in \mathbb{R}$ and for every function $g$, we have

$$I_{z_{[2n]}}(\mathbf{V}; \mathbf{U}_i) \geq \lambda \mathbb{E}_{\mathbf{V}, \mathbf{U}_i | \mathbf{Z}_{[2n]} = z_{[2n]}}[g(\mathbf{V}, \mathbf{U}_i, z_{[2n]})] - \log \mathbb{E}_{\overline{\mathbf{V}}, \overline{\mathbf{U}}_i | \mathbf{Z}_{[2n]} = z_{[2n]}}[e^{\lambda g(\overline{\mathbf{V}}, \overline{\mathbf{U}}_i, z_{[2n]})}]. \tag{18}$$

Next, let $g(\mathbf{V}, \mathbf{U}_i, z_{[2n]}) = \mathbb{1}_{\{y^{\mathbf{U}_i}=y\}} h(\mathbf{V}, z_i^{\mathbf{U}_i}) - \mathbb{1}_{\{y^{-\mathbf{U}_i}=y\}} h(\mathbf{V}, z_i^{-\mathbf{U}_i})$. It is easy to see that $g(\overline{\mathbf{V}}, \overline{\mathbf{U}}_i, z_{[2n]})$ can be rewritten as follows:

$$g(\overline{\mathbf{V}}, \overline{\mathbf{U}}_i, z_{[2n]}) = \overline{\mathbf{U}}_i(\mathbb{1}_{\{y_i^-=y\}} h(\overline{\mathbf{V}}, z_i^-) - \mathbb{1}_{\{y_i^+=y\}} h(\overline{\mathbf{V}}, z_i^+)). \tag{19}$$

Thus, we have

$$\log \mathbb{E}_{\overline{\mathbf{V}}, \overline{\mathbf{U}}_i | \mathbf{Z}_{[2n]}=z_{[2n]}}[e^{\lambda g(\overline{\mathbf{V}}, \overline{\mathbf{U}}_i, z_{[2n]})}] = \log \mathbb{E}_{\overline{\mathbf{V}}, \overline{\mathbf{U}}_i | \mathbf{Z}_{[2n]}=z_{[2n]}}[e^{\lambda \overline{\mathbf{U}}_i(\mathbb{1}_{\{y_i^-=y\}} h(\overline{\mathbf{V}}, z_i^-) - \mathbb{1}_{\{y_i^+=y\}} h(\overline{\mathbf{V}}, z_i^+))}]. \tag{20}$$

Note that $\mathbb{E}_{\overline{\mathbf{U}}_i}[\overline{\mathbf{U}}_i(\mathbb{1}_{\{y_i^-=y\}} h(\overline{\mathbf{V}}, z_i^-) - \mathbb{1}_{\{y_i^+=y\}} h(\overline{\mathbf{V}}, z_i^+))] = 0$ and $\overline{\mathbf{U}}_i \in \{-1, +1\}$. Thus, using Hoeffding's Lemma, we have

$$\log \mathbb{E}_{\overline{\mathbf{V}}, \overline{\mathbf{U}}_i | \mathbf{Z}_{[2n]}=z_{[2n]}}[e^{\lambda g(\overline{\mathbf{V}}, \overline{\mathbf{U}}_i, z_{[2n]})}] \leq \log \mathbb{E}_{\overline{\mathbf{V}} | \mathbf{Z}_{[2n]}=z_{[2n]}}[e^{\frac{\lambda^2}{2}\left(\mathbb{1}_{\{y_i^-=y\}} h(\overline{\mathbf{V}}, z_i^-) - \mathbb{1}_{\{y_i^+=y\}} h(\overline{\mathbf{V}}, z_i^+)\right)^2}]. \tag{21}$$

Next, as $h \in [0, 1]$, $\left| \mathbb{1}_{\{y_i^-=y\}} h(\overline{\mathbf{V}}, z_i^-) - \mathbb{1}_{\{y_i^+=y\}} h(\overline{\mathbf{V}}, z_i^+) \right| \leq \max(\mathbb{1}_{\{y_i^-=y\}}, \mathbb{1}_{\{y_i^+=y\}})$. Thus,

$$\log \mathbb{E}_{\overline{\mathbf{V}}, \overline{\mathbf{U}}_i | \mathbf{Z}_{[2n]}=z_{[2n]}}[e^{\lambda g(\overline{\mathbf{V}}, \overline{\mathbf{U}}_i, z_{[2n]})}] \leq \frac{\lambda^2}{2} \max(\mathbb{1}_{\{y_i^-=y\}}, \mathbb{1}_{\{y_i^+=y\}})^2 = \frac{\lambda^2}{2} \max(\mathbb{1}_{\{y_i^-=y\}}, \mathbb{1}_{\{y_i^+=y\}}). \tag{22}$$

Replacing in equation 18, we have

$$I_{z_{[2n]}}(\mathbf{V}; \mathbf{U}_i) \geq \lambda \mathbb{E}_{\mathbf{V}, \mathbf{U}_i | \mathbf{Z}_{[2n]}=z_{[2n]}}[\mathbb{1}_{y^{\mathbf{U}_i}=y} h(\mathbf{V}, z_i^{\mathbf{U}_i}) - \mathbb{1}_{\{y^{-\mathbf{U}_i}=y\}} h(\mathbf{W}, z_i^{-\mathbf{U}_i})]$$
$$- \frac{\lambda^2}{2} \max(\mathbb{1}_{\{y_i^-=y\}}, \mathbb{1}_{\{y_i^+=y\}}). \tag{23}$$

So, $\forall \lambda \in \mathbb{R}$

$$\frac{\lambda^2}{2} \max(\mathbb{1}_{\{y_i^-=y\}}, \mathbb{1}_{\{y_i^+=y\}}) - \lambda \mathbb{E}_{\mathbf{V}; \mathbf{U}_i | \mathbf{Z}_{[2n]}=z_{[2n]}}[g(\mathbf{V}, \mathbf{U}_i, z_{[2n]})] + I_{z_{[2n]}}(\mathbf{V}; \mathbf{U}_i) \geq 0. \tag{24}$$

The equation 24 is a non-negative parabola with respect to $\lambda$. Thus its discriminant must be non-positive. This implies

$$\mathbb{E}_{\mathbf{V}; \mathbf{U}_i | \mathbf{Z}_{[2n]}=z_{[2n]}}[g(\mathbf{V}, \mathbf{U}_i, z_{[2n]})] \leq \sqrt{2 \max(\mathbb{1}_{\{y_i^-=y\}}, \mathbb{1}_{\{y_i^+=y\}}) I_{z_{[2n]}}(\mathbf{V}; \mathbf{U}_i)}. \tag{25}$$

$\square$

### B.1 PROOF OF THEOREM 1

**Theorem 1** (restated) For $y \in \mathcal{Y}$, assume Assumption 1 holds and the loss $\ell(\overline{\mathbf{W}}, \overline{\mathbf{X}}, y)$ is $\sigma_y$ subgaussian under $P_{\overline{\mathbf{W}}} \otimes P_{\overline{\mathbf{X}}|\overline{\mathbf{Y}}=y}$, then the class-generalization error of class $y$ in Definition 1 can be bounded as:

$$|\overline{\mathrm{gen}_y}(P_{\mathbf{X},\mathbf{Y}}, P_{\mathbf{W}|\mathbf{S}})| \leq \sqrt{2\sigma_y^2 D(P_{\mathbf{W},\mathbf{X}|\mathbf{Y}=y} || P_{\mathbf{W}} \otimes P_{\mathbf{X}|\mathbf{Y}=y})}. \tag{26}$$

*Proof.* From lemma 1, we have

$$\overline{\mathrm{gen}_y}(P_{\mathbf{X},\mathbf{Y}}, P_{\mathbf{W}|\mathbf{S}}) = \mathbb{E}_{P_{\overline{\mathbf{W}}} \otimes P_{\overline{\mathbf{X}}|\overline{\mathbf{Y}}=y}}[\ell(\overline{\mathbf{W}}, \overline{\mathbf{X}}, y)] - \mathbb{E}_{P_{\mathbf{W},\mathbf{X}|\mathbf{Y}=y}}[\ell(\mathbf{W}, \mathbf{X}, y)]. \tag{27}$$

Using the Donsker–Varadhan variational representation of the relative entropy, we have

$$D(P_{\mathbf{W},\mathbf{X}|\mathbf{Y}=y} || P_{\mathbf{W}} \otimes P_{\mathbf{X}|\mathbf{Y}=y}) \geq \mathbb{E}_{P_{\mathbf{W},\mathbf{X}|\mathbf{Y}=y}}[\lambda \ell(\mathbf{W}, \mathbf{X}, y)]$$
$$- \log \mathbb{E}_{P_{\overline{\mathbf{W}}} \otimes P_{\overline{\mathbf{X}}|\overline{\mathbf{Y}}=y}}[e^{\lambda \ell(\overline{\mathbf{W}}, \overline{\mathbf{X}}, y)}], \forall \lambda \in \mathbb{R}. \tag{28}$$

On the other hand, we have:

$$\log \mathbb{E}_{P_{\overline{\mathbf{W}}} \otimes P_{\overline{\mathbf{X}}|\overline{\mathbf{Y}}=y}}\left[e^{\lambda \ell(\overline{\mathbf{W}}, \overline{\mathbf{X}}, y) - \lambda \mathbb{E}[\ell(\overline{\mathbf{W}}, \overline{\mathbf{X}}, y)]}\right]$$
$$= \log \mathbb{E}_{P_{\overline{\mathbf{W}}} \otimes P_{\overline{\mathbf{X}}|\overline{\mathbf{Y}}=y}}\left[e^{\lambda \ell(\overline{\mathbf{W}}, \overline{\mathbf{X}}, y)} e^{-\lambda \mathbb{E}[\ell(\overline{\mathbf{W}}, \overline{\mathbf{X}}, y)])}\right]$$
$$= \log \mathbb{E}_{P_{\overline{\mathbf{W}}} \otimes P_{\overline{\mathbf{X}}|\overline{\mathbf{Y}}=y}}[e^{\lambda \ell(\overline{\mathbf{W}}, \overline{\mathbf{X}}, y)}] - \lambda \mathbb{E}_{P_{\overline{\mathbf{W}}} \otimes P_{\overline{\mathbf{X}}|\overline{\mathbf{Y}}=y}}[\ell(\overline{\mathbf{W}}, \overline{\mathbf{X}}, y)].$$

Using the sub-gaussian assumption, we have

$$\log \mathbb{E}_{P_{\overline{\mathbf{W}}} \otimes P_{\overline{\mathbf{X}}|\overline{\mathbf{Y}}=y}}[e^{\lambda \ell(\overline{\mathbf{W}}, \overline{\mathbf{X}}, y)}] \leq \lambda \mathbb{E}_{P_{\overline{\mathbf{W}}} \otimes P_{\overline{\mathbf{X}}|\overline{\mathbf{Y}}=y}}(\ell(\overline{\mathbf{W}}, \overline{\mathbf{X}}, y)) + \frac{\lambda^2 \sigma_y^2}{2}. \tag{29}$$

By replacing in equation 28, we have

$$D(P_{\mathbf{W}, \mathbf{X}|\mathbf{Y}=y}||P_{\mathbf{W}} \otimes P_{\mathbf{X}|\mathbf{Y}=y}) \geq \lambda\big(\mathbb{E}_{P_{\mathbf{W}, \mathbf{X}|\mathbf{Y}=y}}[\ell(\mathbf{W}, \mathbf{X}, y)] -$$

$$\mathbb{E}_{P_{\overline{\mathbf{W}}} \otimes P_{\overline{\mathbf{X}}|\overline{\mathbf{Y}}=y}}[\ell(\overline{\mathbf{W}}, \overline{\mathbf{X}}, y)]\big) - \frac{\lambda^2 \sigma_y^2}{2}. \tag{30}$$

Thus,

$$D(P_{\mathbf{W}, \mathbf{X}|\mathbf{Y}=y}||P_{\mathbf{W}} \otimes P_{\mathbf{X}|\mathbf{Y}=y}) - \lambda(\mathbb{E}_{P_{\mathbf{W}, \mathbf{X}|\mathbf{Y}=y}}[\ell(\mathbf{W}, \mathbf{X}, y)] - \mathbb{E}_{P_{\overline{\mathbf{W}}} \otimes P_{\overline{\mathbf{X}}|\overline{\mathbf{Y}}=y}}[\ell(\overline{\mathbf{W}}, \overline{\mathbf{X}}, y)])$$

$$+ \lambda^2 \sigma_y^2 \geq 0, \forall \lambda \in \mathbb{R}. \tag{31}$$

Equation equation 31 is a non-negative parabola with respect to $\lambda$, which implies its discriminant must be non-positive. Thus,

$$|\mathbb{E}_{P_{\mathbf{W}, \mathbf{X}|\mathbf{Y}=y}}[\ell(\mathbf{W}, \mathbf{X}, y)] - \mathbb{E}_{P_{\overline{\mathbf{W}}} \otimes P_{\overline{\mathbf{X}}|\overline{\mathbf{Y}}=y}}[\ell(\overline{\mathbf{W}}, \overline{\mathbf{X}}, y)]| \leq \sqrt{2\sigma_y^2 D(P_{\mathbf{W}, \mathbf{X}|\mathbf{Y}=y}||P_{\mathbf{W}} \otimes P_{\mathbf{X}|\mathbf{Y}=y})}. \tag{32}$$

This completes the proof. $\square$

## B.2 PROOF OF THEOREM 2

**Theorem 2** (restated) Assume that the loss $\ell(w, x, y) \in [0, 1]$ is bounded, then the class-generalization error for class $y$ in Definition 2 can be bounded as

$$|\overline{\mathrm{gen}_y}(P_{\mathbf{X}, \mathbf{Y}}, P_{\mathbf{W}|\mathbf{S}})| \leq \mathbb{E}_{\mathbf{Z}_{[2n]}}\Big[\frac{1}{n^y} \sum_{i=1}^{n} \sqrt{2 \max(\mathbb{1}_{\{\mathbf{Y}_i^-=y\}}, \mathbb{1}_{\{\mathbf{Y}_i^+=y\}}) I_{\mathbf{Z}_{[2n]}}(\mathbf{W}; \mathbf{U}_i)}\Big]. \tag{33}$$

*Proof.* Using Lemma 3 with $\mathbf{V} = \mathbf{W}$ and $h(\mathbf{V}, z) = \ell(\mathbf{W}, z)$ in, we have

$$\mathbb{E}_{\mathbf{W}; \mathbf{U}_i|\mathbf{Z}_{[2n]}=z_{[2n]}}[g(\mathbf{W}, \mathbf{U}_i, z_{[2n]})] \leq \sqrt{2 \max(\mathbb{1}_{\{y_i^-=y\}}, \mathbb{1}_{\{y_i^+=y\}}) I_{z_{[2n]}}(\mathbf{W}; \mathbf{U}_i)}, \tag{34}$$

where $g(\mathbf{W}, \mathbf{U}_i, z_{[2n]}) = \mathbb{1}_{\{y^{\mathbf{U}_i}=y\}} \ell(\mathbf{W}, z_i^{\mathbf{U}_i}) - \mathbb{1}_{\{y^{-\mathbf{U}_i}=y\}} \ell(\mathbf{W}, z_i^{-\mathbf{U}_i})$. Thus, by summing over the different terms in Definition 2 and taking expectation over $\mathbf{Z}_{[2n]}$,

$$|\overline{\mathrm{gen}_y}(P_{\mathbf{X}, \mathbf{Y}}, P_{\mathbf{W}|\mathbf{S}})| \leq \mathbb{E}_{\mathbf{Z}_{[2n]}}\Big[\frac{1}{n^y} \sum_{i=1}^{n} \sqrt{2 \max(\mathbb{1}_{\{\mathbf{Y}_i^-=y\}}, \mathbb{1}_{\{\mathbf{Y}_i^+=y\}}) I_{\mathbf{Z}_{[2n]}}(\mathbf{W}; \mathbf{U}_i)}\Big]. \tag{35}$$

$\square$

## B.3 PROOF OF THEOREM 3

**Theorem 3** (restated) Assume that the loss $\ell(\mathbf{W}, \mathbf{X}, y) \in [0, 1]$. The class-generalization error of class $y$, as defined in 2, can be bounded as follows:

$$|\overline{\mathrm{gen}_y}(P_{\mathbf{X}, \mathbf{Y}}, P_{\mathbf{W}|\mathbf{S}})| \leq \mathbb{E}_{\mathbf{Z}_{[2n]}}\Big[\frac{1}{n^y} \sum_{i=1}^{n} \sqrt{2 \max(\mathbb{1}_{\{\mathbf{Y}_i^-=y\}}, \mathbb{1}_{\{\mathbf{Y}_i^+=y\}}) I_{\mathbf{Z}_{[2n]}}(f_{\mathbf{W}}(\mathbf{X}_i^\pm); \mathbf{U}_i)}\Big]. \tag{36}$$

*Proof.* Similar to the proof of Theorem 2. Using Lemma 3 with $\mathbf{V} = f_{\mathbf{W}}(x_i^\pm)$ and $h(\mathbf{V}, z_i) = \ell(f_{\mathbf{W}}(x_i), y_i)$ in, we have

$$\mathbb{E}_{f_{\mathbf{W}}(x_i^\pm); \mathbf{U}_i|\mathbf{Z}_{[2n]}=z_{[2n]}}[g(f_{\mathbf{W}}(\mathbf{X}_i), \mathbf{U}_i, z_{[2n]})] \leq \sqrt{2 \max(\mathbb{1}_{\{y_i^-=y\}}, \mathbb{1}_{\{y_i^+=y\}}) I_{z_{[2n]}}(f_{\mathbf{W}}(x_i^\pm); \mathbf{U}_i)}. \tag{37}$$

Thus taking expectation with respect to $\mathbf{Z}_{[2n]}$ yields the desired result

$$|\overline{\mathrm{gen}_y}(P_{\mathbf{X}, \mathbf{Y}}, P_{\mathbf{W}|\mathbf{S}})| \leq \mathbb{E}_{\mathbf{Z}_{[2n]}}\Big[\frac{1}{n^y} \sum_{i=1}^{n} \sqrt{2 \max(\mathbb{1}_{\{\mathbf{Y}_i^-=y\}}, \mathbb{1}_{\{\mathbf{Y}_i^+=y\}}) I_{\mathbf{Z}_{[2n]}}(f_{\mathbf{W}}(\mathbf{X}_i^\pm); \mathbf{U}_i)}\Big]. \tag{38}$$

$\square$

### B.4 EXTRA BOUND OF CLASS-GENERALIZATION ERROR USING THE LOSS PAIR $\mathbf{L}_i^{\pm}$:

**Theorem 8.** *(class-e-CMI) Assume that the loss $\ell(\hat{y}, y) \in [0, 1]$. The class-generalization error of class $y$, as defined in 2, can be bounded as follows:*

$$|\overline{\mathrm{gen}_y}(P_{\mathbf{X},\mathbf{Y}}, P_{\mathbf{W}|\mathbf{S}})| \leq \mathbb{E}_{\mathbf{Z}_{[2n]}}\left[\frac{1}{n^y}\sum_{i=1}^{n}\sqrt{2\max(\mathbb{1}_{\{\mathbf{Y}_i^-=y\}}, \mathbb{1}_{\{\mathbf{Y}_i^+=y\}})I_{\mathbf{Z}_{[2n]}}(\mathbf{L}_i^{\pm}; \mathbf{U}_i)}\right]. \quad (39)$$

*Proof.* Similar to the proof of Theorems 2 and 3. Using Lemma 3 with $\mathbf{V} = \mathbf{L}_i^{\pm}$ and $h(\mathbf{V}, z_i) = \mathbf{L}_i$ in, we have

$$\mathbb{E}_{\mathbf{L}_i^{\pm};\mathbf{U}_i|\mathbf{Z}_{[2n]}=z_{[2n]}}[g(f_{\mathbf{W}}(\mathbf{X}_i), \mathbf{U}_i, z_{[2n]})] \leq \sqrt{2\max(\mathbb{1}_{\{y_i^-=y\}}, \mathbb{1}_{\{y_i^+=y\}})I_{z_{[2n]}}(\mathbf{L}_i^{\pm}; \mathbf{U}_i)}. \quad (40)$$

Thus taking expectation with respect to $\mathbf{Z}_{[2n]}$ yields the desired result

$$|\overline{\mathrm{gen}_y}(P_{\mathbf{X},\mathbf{Y}}, P_{\mathbf{W}|\mathbf{S}})| \leq \mathbb{E}_{\mathbf{Z}_{[2n]}}\left[\frac{1}{n^y}\sum_{i=1}^{n}\sqrt{2\max(\mathbb{1}_{\{\mathbf{Y}_i^-=y\}}, \mathbb{1}_{\{\mathbf{Y}_i^+=y\}})I_{\mathbf{Z}_{[2n]}}(\mathbf{L}_i^{\pm}; \mathbf{U}_i)}\right]. \quad (41)$$

$\square$

### B.5 PROOF OF THEOREM 4

**Theorem 4** (restated) ($\Delta_y L$-CMI) $\Delta_y\mathbf{L}_i \triangleq \mathbb{1}_{\{y_i^-=y\}}\ell(f_{\mathbf{W}}(\mathbf{X}_i)^-, y_i^-) - \mathbb{1}_{\{y_i^+=y\}}\ell(f_{\mathbf{W}}(\mathbf{X}_i)^+, y_i^+)$. Assume that the loss $\ell(\hat{y}, y) \in [0, 1]$. The class-generalization error of class $y$, as defined in 2, can be bounded as follows:

$$|\overline{\mathrm{gen}_y}(P_{\mathbf{X},\mathbf{Y}}, P_{\mathbf{W}|\mathbf{S}})| \leq \mathbb{E}_{\mathbf{Z}_{[2n]}}\left[\frac{1}{n^y}\sum_{i=1}^{n}\sqrt{2I_{\mathbf{Z}_{[2n]}}(\Delta_y\mathbf{L}_i; \mathbf{U}_i)}\right]. \quad (42)$$

*Proof.* First, we notice that for a fixed realization $z_{[2n]}$, $\mathbb{1}_{\{y^{\mathbf{U}_i}=y\}}\ell(\mathbf{W}, z_i^{\mathbf{U}_i}) - \mathbb{1}_{\{y^{-\mathbf{U}_i}=y\}}\ell(\mathbf{W}, z_i^{-\mathbf{U}_i}) = \mathbf{U}_i(\mathbb{1}_{\{y_i^-=y\}}\ell(\mathbf{W}, z_i^-) - \mathbb{1}_{\{y_i^+=y\}}\ell(\mathbf{W}, z_i^+)) = \mathbf{U}_i\Delta_y\mathbf{L}_i$.

Next, let $(\overline{\Delta_y\mathbf{L}}_i, \overline{\mathbf{U}}_i)$ be an independent copy of $(\Delta_y\mathbf{L}_i; \mathbf{U}_i)$. Using the Donsker–Varadhan variational representation of KL divergence, we have $\forall \lambda \in \mathbb{R}$ and for every function $g$

$$I_{z_{[2n]}}(\Delta_y\mathbf{L}_i; \mathbf{U}_i) \geq \lambda\mathbb{E}_{\Delta_y\mathbf{L}_i, \mathbf{U}_i|\mathbf{Z}_{[2n]}=z_{[2n]}}[g(\Delta_y\mathbf{L}_i, \mathbf{U}_i, z_{[2n]})]$$
$$- \log\mathbb{E}_{\overline{\Delta_y\mathbf{L}}_i, \overline{\mathbf{U}}_i|\mathbf{Z}_{[2n]}=z_{[2n]}}[e^{\lambda g(\overline{\Delta_y\mathbf{L}}_i, \overline{\mathbf{U}}_i, z_{[2n]})}]. \quad (43)$$

Next, let $g(\Delta_y\mathbf{L}_i, \mathbf{U}_i, z_{[2n]}) = \mathbf{U}_i\Delta_y\mathbf{L}_i$. Thus, we have

$$\log\mathbb{E}_{\overline{\Delta_y\mathbf{L}}_i, \overline{\mathbf{U}}_i|\mathbf{Z}_{[2n]}=z_{[2n]}}[e^{\lambda g(\overline{\Delta_y\mathbf{L}}_i, \overline{\mathbf{U}}_i, z_{[2n]})}] = \log\mathbb{E}_{\overline{\Delta_y\mathbf{L}}_i, \overline{\mathbf{U}}_i|\mathbf{Z}_{[2n]}=z_{[2n]}}[e^{\lambda\overline{\mathbf{U}}_i\overline{\Delta_y\mathbf{L}}_i}]. \quad (44)$$

We have $\mathbb{E}_{\overline{\mathbf{U}}_i}[\overline{\mathbf{U}}_i\overline{\Delta_y\mathbf{L}}_i] = 0$ and $\overline{\mathbf{U}}_i \in \{-1, +1\}$. Thus, using Hoeffding's Lemma, we have

$$\log\mathbb{E}_{\overline{\mathbf{L}}_i^{\pm}, \overline{\mathbf{U}}_i|\mathbf{Z}_{[2n]}=z_{[2n]}}[e^{\lambda g(\overline{\Delta_y\mathbf{L}}_i, \overline{\mathbf{U}}_i, z_{[2n]})}] \leq \log\mathbb{E}_{\overline{\Delta_y\mathbf{L}}_i|\mathbf{Z}_{[2n]}=z_{[2n]}}[e^{\frac{\lambda^2}{2}\overline{\Delta_y\mathbf{L}}_i^2}]. \quad (45)$$

Next, as $\ell \in [0, 1]$, it follows that $\Delta_y\mathbf{L}_i \in [-1, 1]$. Thus, $|\overline{\Delta_y\mathbf{L}}_i| \leq 1$. Thus,

$$\log\mathbb{E}_{\overline{\Delta_y\mathbf{L}}_i, \overline{\mathbf{U}}_i|\mathbf{Z}_{[2n]}=z_{[2n]}}[e^{\lambda g(\overline{\Delta_y\mathbf{L}}_i, \overline{\mathbf{U}}_i, z_{[2n]})}] \leq \frac{\lambda^2}{2}. \quad (46)$$

Replacing in equation 43, we have

$$I_{z_{[2n]}}(\Delta_y\mathbf{L}_i; \mathbf{U}_i) \geq \lambda\mathbb{E}_{\Delta_y\mathbf{L}_i, \mathbf{U}_i|\mathbf{Z}_{[2n]}=z_{[2n]}}[\mathbf{U}_i\Delta_y\mathbf{L}_i] - \frac{\lambda^2}{2}. \quad (47)$$

So $\forall \lambda \in \mathbb{R}$

$$\frac{\lambda^2}{2} - \lambda \mathbb{E}_{\Delta_y \mathbf{L}_i; \mathbf{U}_i | \mathbf{Z}_{[2n]} = z_{[2n]}} [g(\Delta_y \mathbf{L}_i, \mathbf{U}_i, z_{[2n]})] + I_{z_{[2n]}}(\Delta_y \mathbf{L}_i; \mathbf{U}_i) \geq 0. \tag{48}$$

The equation 48 is a non-negative parabola with respect to $\lambda$. Thus, its discriminant must be non-positive. This implies

$$\mathbb{E}_{\Delta_y \mathbf{L}_i; \mathbf{U}_i | \mathbf{Z}_{[2n]} = z_{[2n]}} [g(f_{\mathbf{W}}(\mathbf{X}_i), \mathbf{U}_i, z_{[2n]})] \leq \sqrt{2 I_{z_{[2n]}}(\Delta_y \mathbf{L}_i; \mathbf{U}_i)}. \tag{49}$$

Thus taking expectation with respect to $\mathbf{Z}_{[2n]}$ yields the desired result

$$|\overline{\mathrm{gen}_y}(P_{\mathbf{X}, \mathbf{Y}}, P_{\mathbf{W}|\mathbf{S}})| \leq \mathbb{E}_{\mathbf{Z}_{[2n]}} \Big[ \frac{1}{n^y} \sum_{i=1}^{n} \sqrt{2 I_{\mathbf{Z}_{[2n]}}(\Delta_y \mathbf{L}_i; \mathbf{U}_i)} \Big]. \tag{50}$$

**Proof that the $\Delta_y L$-CMI is always tighter than the class-$f$-CMI bound in Theorem 3, and the latter is always tighter than the class-CMI bound in Theorem 2:**

Due to the data processing inequality , we have $\mathbf{U} \to \mathbf{W} \to f_{\mathbf{W}}(\mathbf{X}_i^{\pm}) \to \Delta_y \mathbf{L}_i$. It then follows directly that the class-$f$-CMI bound is always tighter than the class-CMI bound. Moreover, we can also show that for a the $\Delta_y L$-CMI is always tighter than the class-$f$-CMI bound. For a fixed $\mathbf{Z}_{[2n]}$, we have 4 different possible cases for each term in the sum:

- i) If $y_i^- \neq y$ and $y_i^+ \neq y$: In this case, $\max(\mathbb{1}_{\{\mathbf{Y}_i^- = y\}}, \mathbb{1}_{\{\mathbf{Y}_i^+ = y\}}) I_{\mathbf{Z}_{[2n]}}(f_{\mathbf{W}}(\mathbf{X}_i^{\pm}); \mathbf{U}_i) = 0 I_{\mathbf{Z}_{[2n]}}(f_{\mathbf{W}}(\mathbf{X}_i^{\pm}); \mathbf{U}_i) = 0$. On the other hand, we have $\Delta_y \mathbf{L}_i = 0$ so $I_{\mathbf{Z}_{[2n]}}(\Delta_y \mathbf{L}_i; \mathbf{U}_i) = I_{\mathbf{Z}_{[2n]}}(0; \mathbf{U}_i) = 0 \leq 0 = \max(\mathbb{1}_{\{\mathbf{Y}_i^- = y\}}, \mathbb{1}_{\{\mathbf{Y}_i^+ = y\}}) I_{\mathbf{Z}_{[2n]}}(f_{\mathbf{W}}(\mathbf{X}_i^{\pm}); \mathbf{U}_i)$

- ii),If $y_i^- = y$ and $y_i^+ = y$: In this case, $\max(\mathbb{1}_{\{\mathbf{Y}_i^- = y\}}, \mathbb{1}_{\{\mathbf{Y}_i^+ = y\}}) I_{\mathbf{Z}_{[2n]}}(f_{\mathbf{W}}(\mathbf{X}_i^{\pm}); \mathbf{U}_i) = I_{\mathbf{Z}_{[2n]}}(f_{\mathbf{W}}(\mathbf{X}_i^{\pm}); \mathbf{U}_i)$. Due to the data processing inequality, $I_{\mathbf{Z}_{[2n]}}(\Delta_y \mathbf{L}_i; \mathbf{U}_i) \leq I_{\mathbf{Z}_{[2n]}}(f_{\mathbf{W}}(\mathbf{X}_i^{\pm}); \mathbf{U}_i) = \max(\mathbb{1}_{\{\mathbf{Y}_i^- = y\}}, \mathbb{1}_{\{\mathbf{Y}_i^+ = y\}}) I_{\mathbf{Z}_{[2n]}}(f_{\mathbf{W}}(\mathbf{X}_i^{\pm}); \mathbf{U}_i)$

- iii) if ,If $y_i^+ \neq y$ and $y_i^- = y$, In this case, $\max(\mathbb{1}_{\{\mathbf{Y}_i^- = y\}}, \mathbb{1}_{\{\mathbf{Y}_i^+ = y\}}) I_{\mathbf{Z}_{[2n]}}(f_{\mathbf{W}}(\mathbf{X}_i^{\pm}); \mathbf{U}_i) = I_{\mathbf{Z}_{[2n]}}(f_{\mathbf{W}}(\mathbf{X}_i^{\pm}); \mathbf{U}_i)$ and $\Delta_y \mathbf{L}_i = \mathbf{L}_i^+$. As $\mathbf{W} \to f_{\mathbf{W}}(\mathbf{X}_i^{\pm}) \to \mathbf{L}_i^+$ is a Markov chain, using the data processing inequality, we have $I_{\mathbf{Z}_{[2n]}}(\mathbf{L}_i^+; \mathbf{U}_i) \leq I_{\mathbf{Z}_{[2n]}}(f_{\mathbf{W}}(\mathbf{X}_i^{\pm}); \mathbf{U}_i)$ and thus $I_{\mathbf{Z}_{[2n]}}(\Delta_y \mathbf{L}_i; \mathbf{U}_i) \leq I_{\mathbf{Z}_{[2n]}}(f_{\mathbf{W}}(\mathbf{X}_i^{\pm}); \mathbf{U}_i)$

- iv) If $y_i^+ = y$ and $y_i^- \neq y$, this corresponds to the same as iii) by changing the + and -.

As for all the different cases, we have

$$I_{\mathbf{Z}_{[2n]}}(\Delta_y \mathbf{L}_i; \mathbf{U}_i) \leq \max(\mathbb{1}_{\{\mathbf{Y}_i^- = y\}}, \mathbb{1}_{\{\mathbf{Y}_i^+ = y\}}) I_{\mathbf{Z}_{[2n]}}(f_{\mathbf{W}}(\mathbf{X}_i^{\pm}); \mathbf{U}_i) \tag{51}$$

$\Delta_y L$-CMI is always tighter than the class-$f$-CMI bound.

$\square$

## C  ADDITIONAL EMPIRICAL RESULTS

### C.1  EXPERIMENT SETUP

Here, we fully describe the experimental setup used in the main body of the paper. We use the exact same setup as in Harutyunyan et al. (2021), where the code is publicly available[2]. For every number of training data $n$, we draw $m_1$ of the random variable $\mathbf{Z}_{[2n]}$, i.e., we select $m_1$ different

---

[2]https://github.com/hrayrhar/f-CMI/tree/master

2n samples from the original dataset of size $h > 2n$. Then, for each $z_{[2n]}$, we draw $m_2$ different train/test splits, i.e., $m_2$ random realizations of $\mathbf{U}$. In total, we have $m_1 m_2$ experiments. We report the mean and standard deviation on the $m_1$ results. For the CIFER10 experiments, we select $m_1 = 2$ and $m_2 = 20$. For its noisy variant, we select $m_1 = 5$ and $m_2 = 15$. For both datasets, we use ResNet50 pre-trained on ImageNet. The training is conducted for 40 epochs using SGD with a learning rate of 0.01 and a batch size of 256.

In practice, $P(\mathbf{Y} = y)$ is unknown and needs to be estimated using the available data. One simple way is to use $\frac{n^y_{Z_{[2n]}}}{2n}$ as an estimate $P(\mathbf{Y} = y)$. Thus, $n^y = nP(\mathbf{Y} = y)$ used in Section 2.2, can be estimated with $\frac{2}{n^y_{Z_{[2n]}}}$. Note that our empirical results validate the proposed bounds for this estimated class-wise generalization error.

## C.2 FULL CLASS-GENERALIZATION ERROR VS. STANDARD GENERALIZATION RESULTS ON CIFAR10

As a supplement to Figure 1 (left), we plot the standard generalization error along with the class-generalization error of all the classes of CIFAR10 in Figure 3. Similar to the observations highlighted in Section 1, we notice the significant variability in generalization performance across different classes.

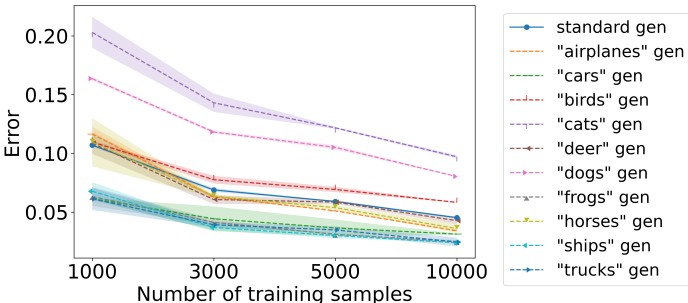

Figure 3: The standard generalization error and the generalization error relative for all classes on CIFAR10 as a function of the number of training data.

## C.3 NUMERICAL RESULTS WITH OUR BOUNDS FOR ALL CLASSES OF CIFAR10 AND ITS NOISY VARIANT

In Figure 4, we present the results of the empirical evaluation of our bounds on all the classes of CIFAR10. Moreover, we generate the scatter plot between the class-generalization error and the class-$f$-CMI bound. We note that similar to the class-$\Delta L_y$ results in Figure 2, our bound scales linearly with the error. The results of noisy CIFAR10, presented in Figure 5, are also consistent with these findings.

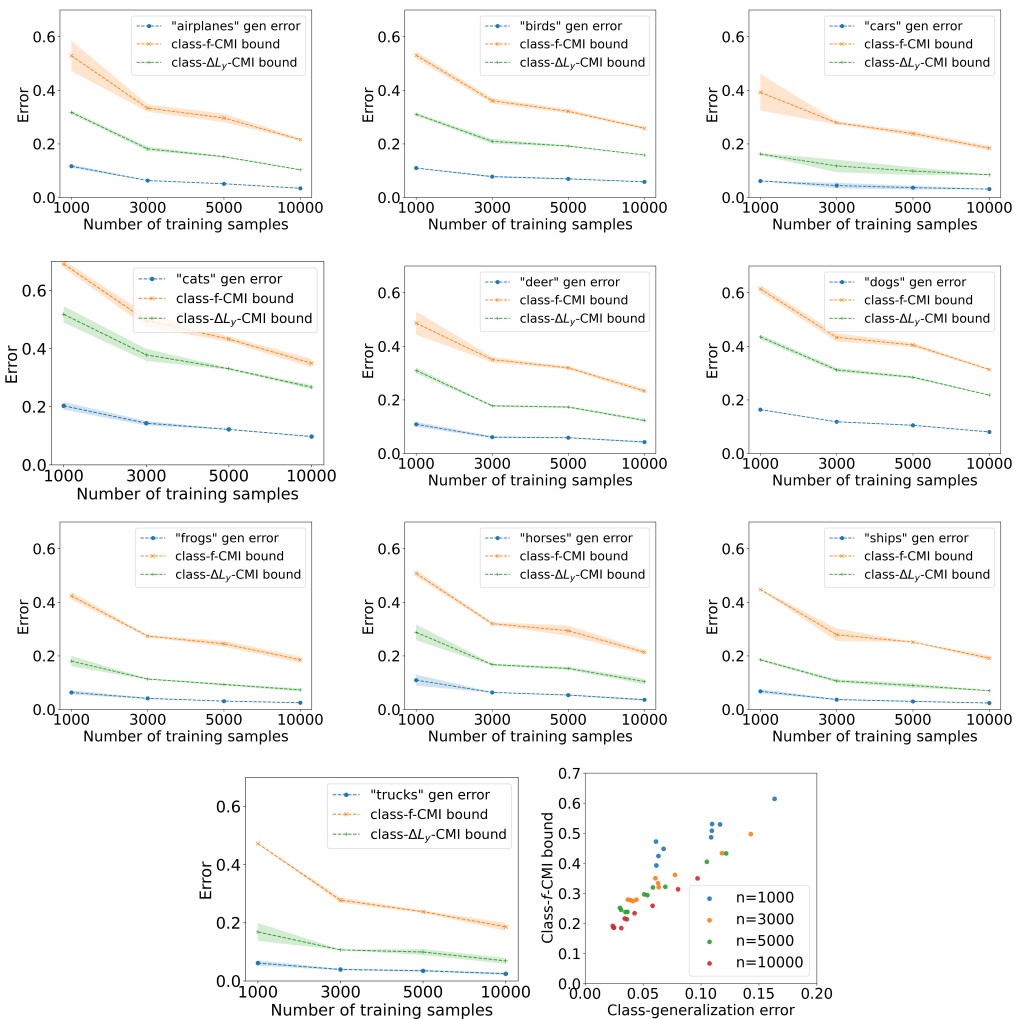

Figure 4: Class-wise generalization on the 10 classes of CIFAR10 and the scatter plot between class-generalization error and the class-$f$-CMI bound in Theorem 3.

### C.4 COROLLARY 1 COMPARED TO THE BOUND IN BU ET AL. (2020)

In the case of standard loss sub-gaussianity assumption, i.e., $\sigma_y = \sigma$ is independent of $y$, it is possible to show that the bound in 1 is tighter than the bound in Bu et al. (2020). This is due to the fact that

$$
\mathbb{E}_{P_Y}\sqrt{2\sigma^2 D(P_{\mathbf{W},\mathbf{X}|\mathbf{Y}}||P_{\mathbf{W}}\otimes P_{\mathbf{X}|\mathbf{Y}})} = \mathbb{E}_{P_Y}\sqrt{2\sigma^2 \mathbb{E}_{P_{\mathbf{W},\mathbf{X}|\mathbf{Y}}}\frac{P_{\mathbf{W},\mathbf{X}|\mathbf{Y}}P_{\mathbf{Y}}}{P_{\mathbf{W}}\otimes P_{\mathbf{X}|\mathbf{Y}}P_{\mathbf{Y}}}}
$$

$$
= \mathbb{E}_{P_Y}\sqrt{2\sigma^2 \mathbb{E}_{P_{\mathbf{W},\mathbf{X}|\mathbf{Y}}}\frac{P_{\mathbf{W},\mathbf{X},\mathbf{Y}}}{P_{\mathbf{W}}\otimes P_{\mathbf{X},\mathbf{Y}}}} \leq \sqrt{2\sigma^2 I(\mathbf{W};\mathbf{Z})} \quad (52)
$$

where the last inequality comes from Jensen's inequality. This shows that class-wise analysis can be used to derive tighter generalization bounds.

### C.5 EXTRA RESULTS: EXPECTED RECALL & SPECIFICITY GENERALIZATION

In the special case of binary classification with the 0-1 loss, the class-generalization errors studied within this paper correspond to generalization in terms of recall and specificity:

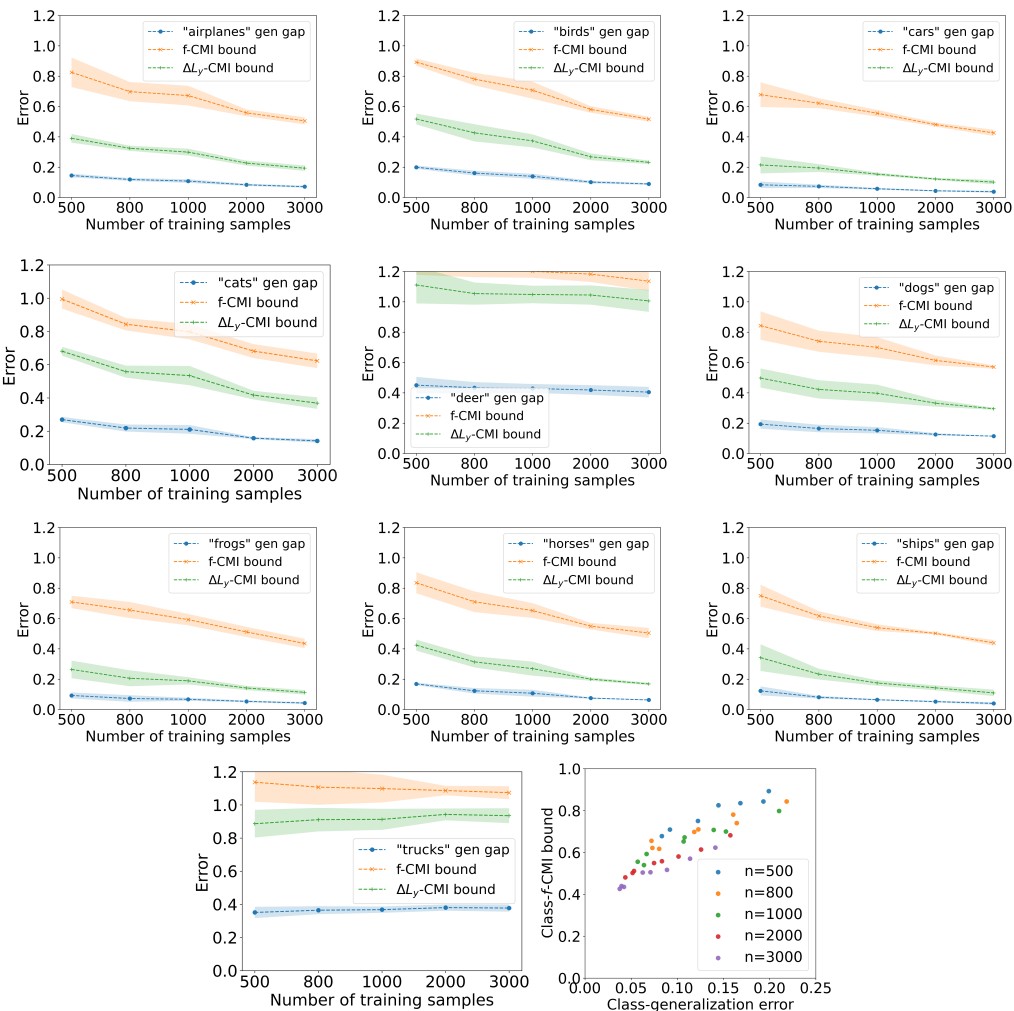

Figure 5: Class-wise generalization on the 10 classes of noisy CIFAR10 (clean validation) and the scatter plot between class-generalization error and the class-$f$-CMI bound in Theorem 3.

$expected\ recall - empirical\ recall = gen_p$, where $p$ is the positive class.
$expected\ specificity - empirical\ specificity = gen_n$, where $n$ is the negative class.

Standard Generalization bounds (Xu & Raginsky, 2017; Harutyunyan et al., 2021) provide theoretical certificates for learning algorithms regarding classification error/accuracy. However, in several ML applications, e.g., an imbalanced binary dataset, accuracy/error are not considered good performance metrics. For example, consider the binary classification problem of detecting a rare cancer type. While analyzing the model's generalization error is important, in this case, we might be more interested in understanding generalization in terms of recall, as that is more critical in this case. Standard theoretical bounds (Neyshabur et al., 2017; Wu et al., 2020; Wang & Mao, 2023) do not provide any insights for such metrics.

The developed tools in this paper can be used to close this gap and allow us to understand generalization for recall and specificity theoretically.

We conduct an experiment of binary MNIST (digit 4 vs. digit 9), similar to Harutyunyan et al. (2021). $m_1$ and $m_2$ discussed in Section C.1 are selected to be $m_1 = 5$ and $m_2 = 30$. Empirical results for this particular case are presented in Figure 7. As can be seen in the Figure, our bounds efficiently estimate the expected recall and specificity errors.

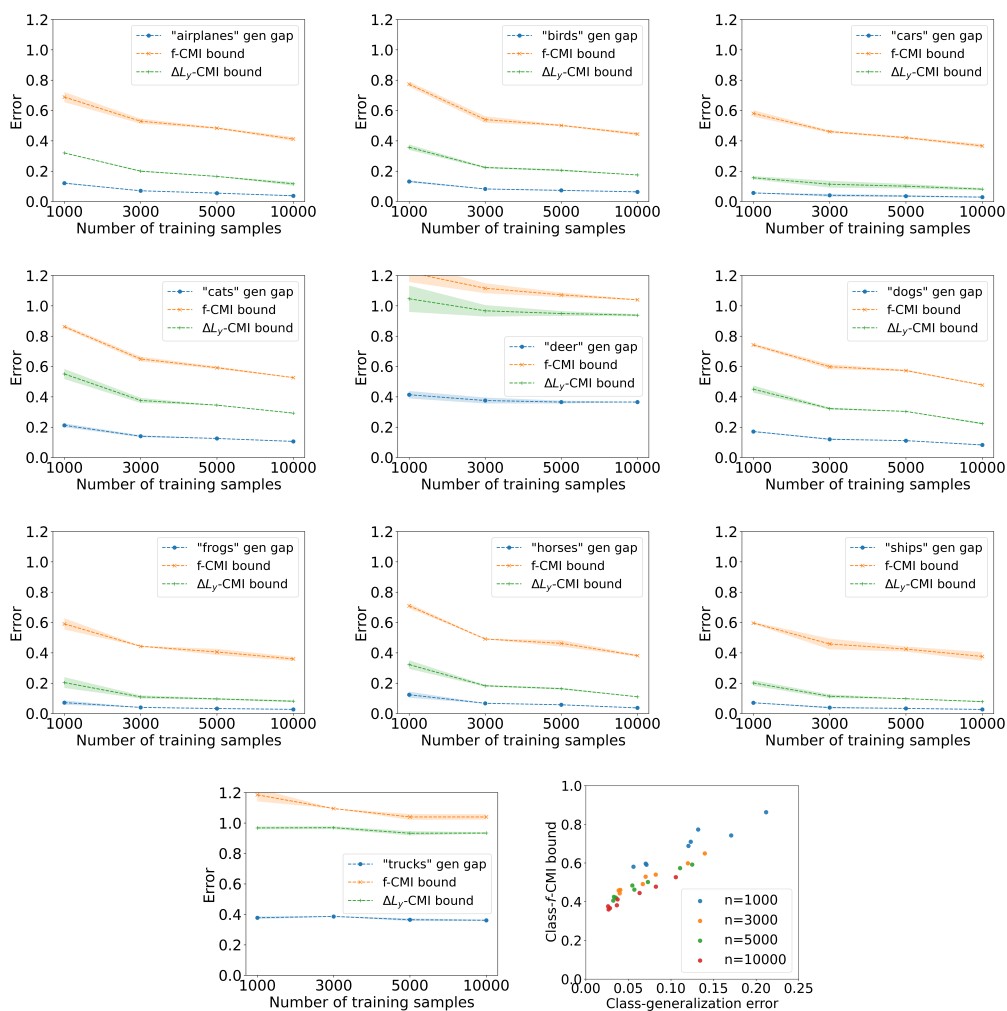

Figure 6: Class-wise generalization on the 10 classes of noisy CIFAR10 (noise added to both train and validation) and the scatter plot between class-generalization error and the class-$f$-CMI bound in Theorem 3.

# D  EXTRA ANALYSIS FOR SECTION 4: OTHER APPLICATIONS

## D.1  FULL DETAILS OF SECTION 4.1: FROM CLASS-GENERALIZATION ERROR TO STANDARD GENERALIZATION ERROR

**Corollary 1** (restated) Assume that for every $y \in \mathcal{Y}$, the loss $\ell(\overline{\mathbf{W}}, \overline{\mathbf{X}}, y)$ is $\sigma_y$ sub-gaussian under $P_{\overline{\mathbf{W}}} \otimes P_{\overline{\mathbf{X}}|\overline{\mathbf{Y}}=y}$, then

$$|\overline{\mathrm{gen}}(P_{\mathbf{X},\mathbf{Y}}, P_{\mathbf{W}|\mathbf{S}})| \leq \frac{1}{n} \sum_{i=1}^{n} \mathbb{E}_Y \sqrt{2\sigma_\mathbf{Y}^2 D(P_{\mathbf{W},\mathbf{X}_i|\mathbf{Y}_i=y} || P_\mathbf{W} \otimes P_{\mathbf{X}_i|\mathbf{Y}_i=y})}. \tag{53}$$

*Proof.* The generalization error can be written as

$$\overline{\mathrm{gen}}(P_{\mathbf{X},\mathbf{Y}}, P_{\mathbf{W}|\mathbf{S}}) = \frac{1}{n} \sum_{i=1}^{n} \left( \mathbb{E}_{\mathbf{W},\overline{\mathbf{Z}}}[\ell(\mathbf{W}, \overline{\mathbf{Z}})] - \mathbb{E}_{\mathbf{W},\mathbf{Z}_i}[\ell(\mathbf{W}, \mathbf{Z}_i)] \right). \tag{54}$$

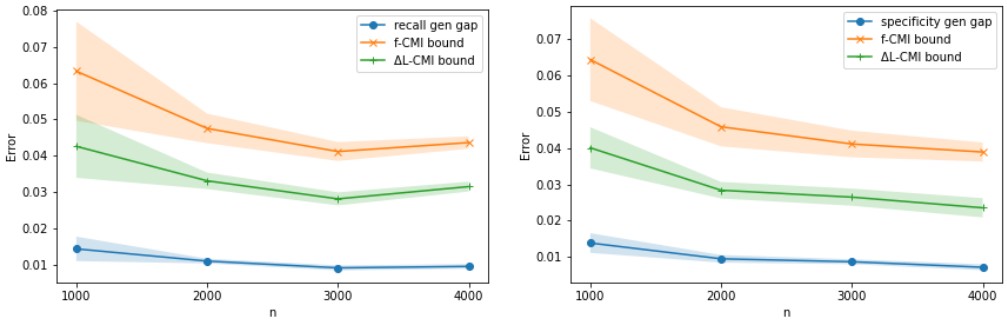

Figure 7: recall generalization error (left) and specificity generalization error (right) for the binary classification 4vs9 from MNIST. The digit 4 is considered the positive class.

As the loss $\ell$ is $\sigma_{\mathbf{Y}}$ sub-gaussian, using Theorem 1, we have

$$\mathbb{E}_{P_{\mathbf{W},\mathbf{X}|\mathbf{Y}=y}}[\ell(\mathbf{W},\mathbf{X},\mathbf{Y})] - \mathbb{E}_{P_{\mathbf{W}}\otimes P_{\mathbf{X}|\mathbf{Y}=y}}[\ell(\overline{\mathbf{W}},\overline{\mathbf{X}},\overline{\mathbf{Y}})] \leq \sqrt{2\sigma_y^2 D(P_{\mathbf{W},\mathbf{X}|\mathbf{Y}=y}||P_{\mathbf{W}}\otimes P_{\mathbf{X}|\mathbf{Y}=y})}.$$
(55)

Taking the expectation over $\mathbf{Y}$ in both sides, we have

$$\mathbb{E}_{P_{\mathbf{W},\mathbf{X},\mathbf{Y}}}[\ell(\mathbf{W},\mathbf{X},\mathbf{Y})] - \mathbb{E}_{P_{\mathbf{W}}\otimes P_{\mathbf{X},\mathbf{Y}}}[\ell(\overline{\mathbf{W}},\overline{\mathbf{X}},\overline{\mathbf{Y}})] \leq \mathbb{E}_Y\sqrt{2\sigma_{\mathbf{Y}}^2 D(P_{\mathbf{W},\mathbf{X}|\mathbf{Y}=y}||P_{\mathbf{W}}\otimes P_{\mathbf{X}|\mathbf{Y}=y})}.$$
(56)

Applying equation 56 on each term of equation 54 for each $\mathbf{Z}_i$ completes the proof.

$\square$

With similar proofs, we can also extend the results in Theorems 2, 3, and 4 into standard generalization bounds by taking the expectation over $y \sim P_{\mathbf{Y}}$. In Corollaries 4, 5, 6, and 2, we provide such an extension of Theorems 2, 3, 8, and 4, respectively.

**Corollary 2**(restated) Assume that the loss $\ell(\hat{y}, y) \in [0, 1]$, then

$$|\overline{\text{gen}}(P_{\mathbf{X},\mathbf{Y}}, P_{\mathbf{W}|\mathbf{S}})| \leq \mathbb{E}_{\mathbf{Y}}\left[\mathbb{E}_{\mathbf{Z}_{[2n]}}\left[\frac{1}{n^{\mathbf{Y}}}\sum_{i=1}^{n}\sqrt{2I_{\mathbf{Z}_{[2n]}}(\Delta_{\mathbf{Y}}\mathbf{L}_i;\mathbf{U}_i)}\right]\right].$$
(57)

**Corollary 4.** *(extra result)* Assume that the loss $\ell(\hat{y}, y) \in [0, 1]$, then

$$|\overline{\text{gen}}(P_{\mathbf{X},\mathbf{Y}}, P_{\mathbf{W}|\mathbf{S}})| \leq \mathbb{E}_{\mathbf{Y}}\left[\mathbb{E}_{\mathbf{Z}_{[2n]}}\left[\frac{1}{n^{\mathbf{Y}}}\sum_{i=1}^{n}\sqrt{2\max(\mathbb{1}_{\mathbf{Y}_i^-=\mathbf{Y}},\mathbb{1}_{\mathbf{Y}_i^+=\mathbf{Y}})I_{\mathbf{Z}_{[2n]}}(\mathbf{W};\mathbf{U}_i)}\right]\right].$$
(58)

**Corollary 5.** *(extra result)* Assume that the loss $\ell(\hat{y}, y) \in [0, 1]$, then

$$|\overline{\text{gen}}(P_{\mathbf{X},\mathbf{Y}}, P_{\mathbf{W}|\mathbf{S}})| \leq \mathbb{E}_{\mathbf{Y}}\left[\mathbb{E}_{\mathbf{Z}_{[2n]}}\left[\frac{1}{n^{\mathbf{Y}}}\sum_{i=1}^{n}\sqrt{2\max(\mathbb{1}_{\mathbf{Y}_i^-=\mathbf{Y}},\mathbb{1}_{\mathbf{Y}_i^+=\mathbf{Y}})I_{\mathbf{Z}_{[2n]}}(f_{\mathbf{W}}(\mathbf{X}_i^{\pm});\mathbf{U}_i)}\right]\right].$$
(59)

**Corollary 6.** *(extra result)* Assume that the loss $\ell(\hat{y}, y) \in [0, 1]$, then

$$|\overline{\text{gen}}(P_{\mathbf{X},\mathbf{Y}}, P_{\mathbf{W}|\mathbf{S}})| \leq \mathbb{E}_{\mathbf{Y}}\left[\mathbb{E}_{\mathbf{Z}_{[2n]}}\left[\frac{1}{n^{\mathbf{Y}}}\sum_{i=1}^{n}\sqrt{2\max(\mathbb{1}_{\mathbf{Y}_i^-=\mathbf{Y}},\mathbb{1}_{\mathbf{Y}_i^+=\mathbf{Y}})I_{\mathbf{Z}_{[2n]}}(\mathbf{L}_i^{\pm};\mathbf{U}_i)}\right]\right].$$
(60)

### D.2 FULL DETAILS OF THE SUB-TASK PROBLEM

Consider a supervised learning problem where the machine learning model $f_{\mathbf{W}}(\cdot)$, parameterized with $w \in \mathcal{W}$, is obtained with a training dataset $S$ consisting of $n$ i.i.d samples $z_i = (x_i, y_i) \in \mathcal{X} \times \mathcal{Y} \triangleq \mathcal{Z}$ generated from distribution $P_{\mathbf{X}\mathbf{Y}}$. The quality of the model with parameter $w$ is evaluated with a loss function $\ell : \mathcal{W} \times \mathcal{Z} \to \mathbb{R}^+$.

For any $w \in \mathcal{W}$, the population risk is defined as follows

$$L_P(w) = \mathbb{E}_{P_{\mathbf{X},\mathbf{Y}}}[\ell(w, \mathbf{X}, \mathbf{Y})]. \tag{61}$$

and the empirical risk is:

$$L_{E_P}(w, S) = \frac{1}{n} \sum_{i=1}^{n} \ell(w, x_i, y_i). \tag{62}$$

Here, we are interested in the subtask problem, which is a special case of distribution shift, i.e., the test performance of the model $w$ is evaluated using a specific subset of classes $\mathcal{A} \subset \mathcal{Y}$ of the source distribution $P_{\mathbf{XY}}$. Thus, the target distribution $Q_{\mathbf{XY}}$ is defined as $Q_{\mathbf{XY}}(x, y) = \frac{P_{\mathbf{XY}}(x,y)\mathbb{1}_{\{y \in \mathcal{A}\}}}{P_{\mathbf{Y}}(y \in \mathcal{A})}$. The population risk on the target domain $Q$ of the subtask problem is

$$L_Q(w) = \mathbb{E}_{Q_{\mathbf{X},\mathbf{Y}}}[\ell(w, \mathbf{X}, \mathbf{Y})]. \tag{63}$$

A learning algorithm can be modeled as a randomized mapping from the training set $S$ onto a model parameter $w \in \mathcal{W}$ according to the conditional distribution $P_{\mathbf{W}|S}$. The expected generalization error on the subtask problem is the difference between the population risk of $Q$ and the empirical risk evaluated using all samples from $S$:

$$\overline{\text{gen}}_{Q,E_P} = \mathbb{E}_{P_{\mathbf{W},\mathbf{S}}}[L_Q(\mathbf{W}) - L_{E_P}(\mathbf{W}, \mathbf{S})], \tag{64}$$

where the expectation is taken over the joint distribution $P_{\mathbf{W},\mathbf{S}} = P_{\mathbf{W}|S} \otimes P_{\mathbf{Z}}^n$.

The generalization error defined above can be decomposed as follows:

$$\overline{\text{gen}}_{Q,E_P} = \mathbb{E}_{P_{\mathbf{W}}}[L_Q(\mathbf{W}) - L_P(\mathbf{W})] + \mathbb{E}_{P_{\mathbf{W},\mathbf{S}}}[L_P(\mathbf{W}) - L_{E_P}(\mathbf{W}, \mathbf{S})]. \tag{65}$$

The first term quantifies the gap of the population risks in two different domains, and the second term is the source domain generalization error. Assuming that loss is $\sigma$-subgaussian under $P_{\mathbf{Z}}$, it is shown in Wang & Mao (2022) that the first term can be bounded using the KL divergence between $P$ and $Q$:

$$\mathbb{E}_{P_{\mathbf{W}}}[L_Q(w) - L_P(w)] \leq \sqrt{2\sigma^2 D(Q\|P)}. \tag{66}$$

The second term can be bounded using the standard mutual information approach in Xu & Raginsky (2017) as

$$\mathbb{E}_{P_{\mathbf{W},\mathbf{S}}}[L_P(\mathbf{W}) - L_{E_P}(\mathbf{W}, \mathbf{S})] \leq \sqrt{2\sigma^2 I(\mathbf{W}; \mathbf{S})}. \tag{67}$$

Thus, the generalization error of the subtask problem can be bounded as follows:

$$\overline{\text{gen}}_{Q,E_P} \leq \sqrt{2\sigma^2 D(Q\|P)} + \sqrt{2\sigma^2 I(\mathbf{W}; \mathbf{S})}. \tag{68}$$

Obtaining tighter generalization error bounds for the subtask problem is straightforward using our class-wise generalization bounds. In fact, the generalization error bound of the subtask can be obtained by taking the expectation of $\mathbf{Y} \sim Q_{\mathbf{Y}}$.

Using Jensen's inequality, we have $|\overline{\text{gen}}_{Q,E_Q}| = |\mathbb{E}_{\mathbf{Y} \sim Q_{\mathbf{Y}}}[\overline{\text{gen}}_{\mathbf{Y}}]| \leq \mathbb{E}_{\mathbf{Y} \sim Q_{\mathbf{Y}}}[|\overline{\text{gen}}_{\mathbf{Y}}|]$. Thus, we can use the results from Section 2 to obtain tighter bounds.

**Theorem 5** (subtask-CMI) (restated) Assume that the loss $\ell(w, x, y) \in [0, 1]$ is bounded, then the subtask generalization error defined in 11 can be bounded as

$$|\overline{\text{gen}}_{Q,E_Q}| \leq \mathbb{E}_{\mathbf{Y} \sim Q_{\mathbf{Y}}}\left[\mathbb{E}_{\mathbf{Z}_{[2n]}}\left[\frac{1}{n^{\mathbf{Y}}} \sum_{i=1}^{n} \sqrt{2\max(\mathbb{1}_{\mathbf{Y}_i^- = \mathbf{Y}}, \mathbb{1}_{\mathbf{Y}_i^+ = \mathbf{Y}}) I_{\mathbf{Z}_{[2n]}}(\mathbf{W}; \mathbf{U}_i)}\right]\right].$$

Similarly, we can extend the result of class-$\Delta L_y$-CMI, with similar proof, as follows:

**Theorem 6** (subtask-$\Delta L_y$-CMI) (restated) Assume that the loss $\ell(w, x, y) \in [0, 1]$ is bounded, Then the subtask generalization error defined in 11 can be bounded as

$$|\overline{\text{gen}}_{Q,E_Q}| \leq \mathbb{E}_{\mathbf{Y} \sim Q_{\mathbf{Y}}}\left[\mathbb{E}_{\mathbf{Z}_{[2n]}}\left[\frac{1}{n^{\mathbf{Y}}} \sum_{i=1}^{n} \sqrt{2 I_{\mathbf{Z}_{[2n]}}(\Delta_{\mathbf{Y}}\mathbf{L}_i; \mathbf{U}_i)}\right]\right].$$

Similarly, we can also extend the result of Theorem 8 to the subtask as follows:

**Theorem 9.** *(subtask-e-CMI) (extra result) Assume that the loss $\ell(w, x, y) \in [0, 1]$ is bounded, then the subtask generalization error defined in 11 can be bounded as*

$$|\overline{\text{gen}}_{Q,E_Q}| \leq \mathbb{E}_{\mathbf{Y} \sim Q_{\mathbf{Y}}}\left[\mathbb{E}_{\mathbf{Z}_{[2n]}}\left[\frac{1}{n^{\mathbf{Y}}} \sum_{i=1}^{n} \sqrt{2\max(\mathbb{1}_{\mathbf{Y}_i^- = \mathbf{Y}}, \mathbb{1}_{\mathbf{Y}_i^+ = \mathbf{Y}}) I_{\mathbf{Z}_{[2n]}}(f_{\mathbf{W}}(\mathbf{X}_i^{\pm}); \mathbf{U}_i)}\right]\right].$$

### D.3 Extra details of Section 4.3: Generalization certificates with sensitive attributes

**Theorem 7** (restated) Given $t \in \mathcal{T}$, assume that the loss $\ell(\mathbf{W}, \mathbf{Z})$ is $\sigma$ sub-gaussian under $P_{\overline{\mathbf{W}}} \otimes P_{\overline{\mathbf{Z}}}$, then the attribute-generalization error of the sub-population $\mathbf{T} = t$, as defined in 3, can be bounded as follows:

$$|\overline{\mathrm{gen}_t}(P_{\mathbf{X},\mathbf{Y}}, P_{\mathbf{W}|\mathbf{S}})| \leq \sqrt{2\sigma^2 D(P_{\mathbf{W}|\mathbf{Z}} \otimes P_{\mathbf{Z}|\mathbf{T}=t} || P_{\mathbf{W}} \otimes P_{\mathbf{Z}|\overline{\mathbf{T}}=t})}. \tag{69}$$

*Proof.* We have

$$\overline{\mathrm{gen}_t}(P_{\mathbf{X},\mathbf{Y}}, P_{\mathbf{W}|\mathbf{S}}) = \mathbb{E}_{P_{\overline{\mathbf{W}}} \otimes P_{\overline{\mathbf{Z}}|\mathbf{T}=t}}[\ell(\overline{\mathbf{W}}, \overline{\mathbf{Z}})] - \mathbb{E}_{P_{\mathbf{W}|\mathbf{Z}} \otimes P_{\mathbf{Z}|\mathbf{T}=t}}[\ell(\mathbf{W}, \mathbf{Z})]. \tag{70}$$

Using the Donsker–Varadhan variational representation of the relative entropy, we have

$$D(P_{\mathbf{W}|\mathbf{Z}} \otimes P_{\mathbf{Z}|\mathbf{T}=t} || P_{\mathbf{W}} \otimes P_{\mathbf{Z}|\mathbf{T}=t}) \geq \mathbb{E}_{P_{\mathbf{W}|\mathbf{Z}} \otimes P_{\mathbf{Z}|\mathbf{T}=t}}[\lambda \ell(\mathbf{W}, \mathbf{Z})]$$
$$- \log \mathbb{E}_{P_{\overline{\mathbf{W}}} \otimes P_{\overline{\mathbf{Z}}|\overline{\mathbf{T}}=t}}[e^{\lambda \ell(\overline{\mathbf{W}}, \overline{\mathbf{Z}})}], \forall \lambda \in \mathbb{R}. \tag{71}$$

On the other hand, we have:

$$\log \mathbb{E}_{P_{\overline{\mathbf{W}}} \otimes P_{\overline{\mathbf{Z}}|\overline{\mathbf{T}}=t}}\left[ e^{\lambda \ell(\overline{\mathbf{W}}, \overline{\mathbf{Z}}) - \lambda \mathbb{E}[\ell(\overline{\mathbf{W}}, \overline{\mathbf{Z}})]} \right]$$
$$= \log \mathbb{E}_{P_{\overline{\mathbf{W}}} \otimes P_{\overline{\mathbf{Z}}|\overline{\mathbf{T}}=t}}\left[ e^{\lambda \ell(\overline{\mathbf{W}}, \overline{\mathbf{Z}})} e^{-\lambda \mathbb{E}[\ell(\overline{\mathbf{W}}, \overline{\mathbf{Z}})])} \right]$$
$$= \log \mathbb{E}_{P_{\overline{\mathbf{W}}} \otimes P_{\overline{\mathbf{Z}}|\overline{\mathbf{T}}=t}}[e^{\lambda \ell(\overline{\mathbf{W}}, \overline{\mathbf{Z}})}] - \lambda \mathbb{E}_{P_{\overline{\mathbf{W}}} \otimes P_{\overline{\mathbf{Z}}|\overline{\mathbf{T}}=t}}[\ell(\overline{\mathbf{W}}, \overline{\mathbf{Z}})].$$

Using the sub-gaussian assumption, we have

$$\log \mathbb{E}_{P_{\overline{\mathbf{W}}} \otimes P_{\overline{\mathbf{Z}}|\overline{\mathbf{T}}=t}}[e^{\lambda \ell(\overline{\mathbf{W}}, \overline{\mathbf{Z}})}] \leq \lambda \mathbb{E}_{P_{\overline{\mathbf{W}}} \otimes P_{\overline{\mathbf{Z}}|\overline{\mathbf{T}}=t}}(\ell(\overline{\mathbf{W}}, \overline{\mathbf{Z}})) + \frac{\lambda^2 \sigma^2}{2}. \tag{72}$$

By replacing in equation 71, we have

$$D(P_{\mathbf{W}|\mathbf{Z}} \otimes P_{\mathbf{Z}|\mathbf{T}=t} || P_{\mathbf{W}} \otimes P_{\mathbf{Z}|\mathbf{T}=t}) \geq \lambda \big( \mathbb{E}_{P_{\mathbf{W}|\mathbf{Z}} \otimes P_{\mathbf{Z}|\mathbf{T}=t}}[\ell(\mathbf{W}, \mathbf{Z})] -$$
$$\mathbb{E}_{P_{\overline{\mathbf{W}}} \otimes P_{\overline{\mathbf{Z}}|\overline{\mathbf{T}}=t}}[\ell(\overline{\mathbf{W}}, \overline{\mathbf{Z}})] \big) - \frac{\lambda^2 \sigma}{2}. \tag{73}$$

Thus, we have:

$$D(P_{\mathbf{W}|\mathbf{Z}} \otimes P_{\mathbf{Z}|\mathbf{T}=t} || P_{\mathbf{W}} \otimes P_{\mathbf{Z}|\mathbf{T}=t}) - \lambda(\mathbb{E}_{P_{\mathbf{W}|\mathbf{Z}} \otimes P_{\mathbf{Z}|\mathbf{T}=t}}[\ell(\mathbf{W}, \mathbf{Z})] - \mathbb{E}_{P_{\overline{\mathbf{W}}} \otimes P_{\overline{\mathbf{Z}}|\overline{\mathbf{T}}=t}}[\ell(\overline{\mathbf{W}}, \overline{\mathbf{Z}})])$$
$$+ \lambda^2 \sigma^2 \geq 0, \forall \lambda \in \mathbb{R}. \tag{74}$$

equation 74 is a non-negative parabola with respect to $\lambda$. Thus, its discriminant must be non-positive. This implies

$$|\mathbb{E}_{P_{\mathbf{W}|\mathbf{Z}} \otimes P_{\mathbf{Z}|\mathbf{T}=t}}[\ell(\mathbf{W}, \mathbf{Z})] - \mathbb{E}_{P_{\overline{\mathbf{W}}} \otimes P_{\overline{\mathbf{Z}}|\overline{\mathbf{T}}=t}}[\ell(\overline{\mathbf{W}}, \overline{\mathbf{Z}})]| \leq \sqrt{2\sigma^2 D(P_{\mathbf{W}|\mathbf{Z}} \otimes P_{\mathbf{Z}|\mathbf{T}=t} || P_{\mathbf{W}} \otimes P_{\mathbf{Z}|\mathbf{T}=t})}. \tag{75}$$

This completes the proof. $\square$

