# OpenReview forum: "Class-Wise Generalization Error: An Information-Theoretic Analysis"
_ICLR.cc/2024/Conference — Submitted to ICLR 2024_

### Official Review · Reviewer_ceHs · 2023-10-24

**Soundness:** 2 fair
**Presentation:** 2 fair
**Contribution:** 2 fair
**Rating:** 5
**Confidence:** 4

**Summary:**

This paper introduces the concept of "class-generalization error," which measures the generalization performance of the algorithm for each individual class. It further applies previous information-theoretic tools to derive various bounds. Empirical studies validate the proposed bounds in different neural networks. Finally, the authors discussed some possible applications of these bounds.

**Strengths:**

1. The motivation to investigate the class-wise generalization error is reasonable.
2. Although not in-depth enough, this paper has related the class-generalization error to several applications.

**Weaknesses:**

### Major Concerns
1. "...provide bounds for the expected generalization over the whole data distribution, which implicitly assumes that the model generalizes uniformly for all the classes."
   * I am afraid that I cannot agree with this statement. Considering the expected generalization error over the whole data distribution means measuring the model **on average** over all the classes, not **uniformly**, especially when the classes are imbalanced.
2. The number of training samples in Figure 1 is the whole sample number of all the classes.
   * To interpret the reason for the different generalization gaps over classes, we need first to consider whether the classes are balanced.
      - In the website of the CIFAR 10 dataset: "...The training batches contain the remaining images in random order, but some training batches **may contain more images from one class than another**. Between them, the training batches contain exactly 5000 images from each class."
      -  So, could the authors also plot the class-wise gap w.r.t the sample numbers for each class?
   * Moreover, introducing label noise (5%) for each class will cause an imbalanced sample number for each class, which means the class-wise sample numbers in the right plot must be different (in terms of the given label not true label). That's why we observe a different trend.
     - How the label noise is introduced? Have you introduced the noise on both train data and test data? If only on the train data, there is a distribution shift. If on both, it's the above-mentioned class imbalance.
3. Assumption 1, Lemma 1, and Theorem 1 seem problematic.
   * The authors assume that the learning algorithm is symmetric w.r.t each training sample. As the paper is motivated, the generalization gap is non-uniform w.r.t each class. How could two samples from different classes be symmetric to the algorithm output of the whole dataset $P_{W|S}$?
   * Could the authors provide a detailed derivation for Lemma 1? I cannot find any proof for this.
   * Based on Assumption 1 and  Lemma 1, Theorem 1 is not related to the sample number $n_y$ and indicates all the samples inside a class contribute equally to the generalization. However, this is counterintuitive, where the samples close to the decision boundary should be more important.
4. The theorems are based on tiny modifications on previous MI and CMI bounds, which are not novel to me.

### Minor Concern
The citation format is incorrect, please pay attention to the use of citet and citep.

**Questions:**

Please see the previous section

---

> ### Author Response · Authors · 2023-11-22
> **Answers to Reviewer ceHs**
>
> Thank you for your review.
>
> >**Q**: "...provide bounds for the expected generalization over the whole data distribution, which implicitly assumes that the model generalizes uniformly for all the classes."
>
> We agree with the Reviewer that ‘uniformly’ can be interpreted solely from the context of imbalanced datasets. In the context of our paper, we use ‘uniformly’ more generally to the noticeable heterogeneity of generalization among different classes. To avoid this misunderstanding, we have replaced the word uniformly in the text accordingly.
>
> >**Q**:  The number of training samples in Figure 1 is the whole sample number of all the classes. To interpret the reason for the different generalization gaps over classes, we need first to consider whether the classes are balanced.
>
> In Figure 1, we use the total training sample and not the samples corresponding to a specific class. In our framework, we resample the entire training data to obtain i.i.d samples generated from the same data distribution P_Z, so that we can study the average class-wise generalization error. For example, the results shown in Figure 1 are the average over multiple random seeds, i.e., average generalization (For CIFAR10 we have 40 experiments, and for the noisy variant 75) and some of these seeds would be more imbalanced than the others, but the randomness is coming from the data distribution P_Z. As CIFAR10 is a balanced dataset, resample would not create significant imbalanced training data, and the total training sample divided by 10 can be interpreted as the number of samples of each class.
>
> We respectfully disagree with the Reviewer that class imbalance is the main cause of the variations in class-wise generalization errors. Although class imbalance might aggravate this phenomenon, such a variation is a general intrinsic property of neural networks and has been observed in multiple literature even with balanced data. For example, Balestriero et al. (2022); Kirichenko et al. (2023) showed that standard data augmentation and regularization techniques, e.g., weight decay and dropout, surprisingly aggravate this class bias yielding in larger class performance variations. In this work, we are interested in studying this phenomenon theoretically and providing insights about it using information-theoretic tools.  This shows that even ‘benign’ techniques such as classic data augmentation and regularization approaches exhibit pronounced class dependencies. In other words, while these methods enhance generalization for specific classes, they can harm the performance of other classes, even under class-balanced cases.
>
> >**Q**: How the label noise is introduced?
>
> In our experiments, we treat label noise as a data augmentation technique, i.e.,  it is added only to the training data. The main point was to show that a slight modification to the learning technique (data augmentation with label noise in this case) dramatically changes the behavior of class-wise generalization. This further motivates the study of this phenomenon, which is the main contribution of this paper.
> Note that even if the noise is added to both train and test, we still observe similar behavior, and similar conclusions can be drawn. We have included these results in the revised manuscript (Figure 6 in Appendix C.6).
>
> >**Q**: Assumption 1, Lemma 1, and Theorem 1 seem problematic.
>
> Note that Assumption 1 states that the learned model W and each sample Z_i have the same joint distribution PW,Z. Here, the index i can be interpreted as the order of samples in the training data, and Assumption 1 is saying that swapping the order of any two i.i.d samples $Z_i$ and $Z_j$ in the training data will not change the joint distribution of P_WZ. This is a standard assumption in the MI setting to derive generalization bounds based on individual samples instead of the whole dataset (similar to the analysis by Bu et al. (2019)). And we have rephrased the interpretation of this assumption in the revision.
>
> Using the symmetry property of the joint distribution (Assumption 1), it follows directly that all the terms in the sum in Definition 1 become identical and we obtain the individual-sample-based expression of the class-wise generalization in Lemma 1.
>
> In Theorem 1, the conditional KL divergence term implicitly depends on the number of samples. Note that W is learned from the entire training data with n samples, and  PW,X|y denotes the joint distribution of W and a single training data (X,y). The generalization error is upper bounded by the KL divergence between this PW,X|y and the distribution PW PX|Y=y, which replaces one training data with its independent copy. Thus, this KL divergence decreases with $n$.
>
> >**Q**: The theorems are based on tiny modifications on previous MI and CMI bounds, which are not novel to me.
>
> We respectfully disagree with the reviewer. Please refer to our general answer for full details regarding the novelty and the technical challenges in our analysis.

---

### Official Review · Reviewer_yBBn · 2023-10-27

**Soundness:** 3 good
**Presentation:** 3 good
**Contribution:** 2 fair
**Rating:** 5
**Confidence:** 4

**Summary:**

Several recent works have provided information-theoretic generalization bounds, which apply in an average case over all classes. This paper takes a more fine-grained approach, and obtains bounds that hold for each specific class. Specifically, the difference between training and population loss for a given class is bounded in terms of an information measure involving training data for that specific class. The bounds are numerically evaluated. Some applications and extensions are discussed, including the sub-task problem and sensitive attributes.

**Strengths:**

The paper gives a clear presentation of the problem at hand, and motivates well why it is interesting to tackle. The numerical evaluation is detailed for the main problem of the paper. Related literature is mostly well-covered. The connections to the sub-task problem and sensitive attributes are intriguing.

**Weaknesses:**

The theoretical machinery is essentially identical to prior work (Xu & Raginsky, Steinke & Zakynthinou, Bu & Veeravalli, Harutyunyan et al, etc). The only difference is in the introduction of the definition of classwise generalization error — after this, all the steps proceed as in these earlier works (some minor differences in the current CMI derivation). It also seems to me that the results can be derived in simpler ways (see questions below). Now, this in itself is not necessarily a weakness, provided that the end product is useful. There are some steps towards establishing this in Section 4, which to me is the most interesting part of the paper, but it does not go quite far enough. For instance, evaluations on particular sub-task problems, or algorithmic design for fairness inspired by the results, could be promising directions for extending these results.

**Questions:**

— It seems as if all of your results follow straightforwardly by the same technique as you use for Theorem 7. Consider a distribution $P_Z$ on a space $\mathcal Z = \mathcal T \times \mathcal X$, where $\mathcal T$ is, for instance, the label space or a sensitive attribute. Then, apply standard techniques for information-theoretic bounds with the joint distribution $P_{W\vert Z}P_{Z\vert T=t}$. With this, you can avoid the machinery with indicator functions used for the CMI derivations.

In the CMI setting, I am aware that $W$ still depends on the parts of the supersample where the labels are not the specified one in the class-generalization error. But since this part of the supersample only appears through $W$, it is marginalized out. In a similar vein, $W$ can depend on any other random variable (e.g., pre-trained), which also becomes marginalized out.

One potential way to exploit this would be to actually use the non-$t$ data to construct a more informed prior. Consider the data-split method in PAC-Bayesian bounds, as in “Tighter PAC-Bayes bounds” by Ambroladze et al or “On The role of Data in PAC-Bayes Bounds” by Dziugaite et al. This can be combined with the CMI approach (or the standard MI bounds), so that the prior is allowed to depend on data from $P_{Z\vert T\neq t}$.

— For the sub-task problem, are the target classes assumed to be known?

— Regarding the worst sub-population performance: isn’t it the case that $\text{gen} = E_T \text{gen}_T \leq \max_t  \text{gen}_t$? It is not clear to me why it needs to be re-written as an information-theoretic bound to establish a statement similar to (15). Furthermore, does this really motivate that the two are closely correlated? The maximum is clearly an upper bound, but I do not see why this indicates that they should be correlated.

***
Minor comments:

— After 1, you require $n_y < n$. Isn’t $n_y\leq n$ fine?

— The term “overfit” seems a bit unclear (compare “benign overfitting”, e.g.). Does this necessarily imply a high population loss?

— I believe the term “supersample” usually refers to the entire set of $2n$ samples. It seems that you use it to only refer to a single pair of samples. Is this correct?

Assumption 1: satisfying -> satisfies; \citep when needed

**Details Of Ethics Concerns:**

N/A (accidentally wrote questions here)

---

> ### Author Response · Authors · 2023-11-22
> **Answers to Reviewer yBBn**
>
> Thank you for your review. In light of our responses to your individual points, please let us know if you have any remaining concerns.
>
> >**Q**:  It seems as if all of your results follow straightforwardly by the same technique as you use for Theorem 7
>
> We respectfully disagree with the reviewer that all bounds can be obtained similarly to Theorem 7. While Theorem 1 and Theorem 7 both share the same core proof and can be ‘combined’ as the reviewer suggested, we don’t see how the same logic can be applied to the supersample (CMI) setting to obtain bounds without the indicator functions. If possible, would the reviewer elaborate more on this? Indeed, the indicators are a fundamental factor in the definition of the class-wise generalization error in the CMI setting and hence it is expected that they appear in the bounds.
>
> >**Q**: One potential way to exploit this would be to actually use the non-y data to construct a more informed prior in PAC-Bayesian bounds similar to Ambroladze et al (2021)
>
> The Reviewer made an interesting possible connection here. Indeed, one potential way to use our analysis is to extend it to PAC-Bayesian bounds and exploit the non-y data in constructing the prior. However, It is beyond the scope of this work, as here we are interested in deriving bounds in the standard MI and CMI settings.
>
> We are appreciative of the Reviewer's valuable suggestion, and we intend to include it in the section on future work, as we believe it is a promising idea worthy of further investigation.
>
> >**Q**: For the sub-task problem, are the target classes assumed to be known?
>
>  Yes. As explained at the start of Section 4.2 and in Appendix D.2, it is assumed that the target domain encompasses a specific known subset of the classes encountered during training in the sub-task problem.
>
> >**Q**: Regarding the worst sub-population performance. It is not clear to me why it needs to be rewritten as an information-theoretic bound
>
> The reviewer made a valid point here. The maximum is an upper bound without the need for an information-theoretic bound. In the revised manuscript, we have removed eq 15 and its corresponding discussion. In the original draft, we tried to draw a connection between our results and the empirical findings of Yang et al. (2023), where it is shown empirically that model performance is highly correlated with the worst sub-population performance. As our bound in eq 15 depends on the population achieving the worst KL divergence, it can be loosely interpreted as a justification for the empirical results of Yang et al. (2023).
>
> >**Q**:  After 1, you require $n_y < n$. $n_y \leq n$ Isn’t fine?
>
> The case $n_y = n$ refers to the case of having only a single label in the data. In this case, the problem is ill-posed as there is only one class in the data (no learning needed for a classification problem).
>
> >**Q**: The term “overfit” seems a bit unclear (compare “benign overfitting”, e.g.).
>
> In the paper, overfitting refers to the classic phenomenon of having a large generalization error, i.e., a large gap between the empirical risk and the population (test) risk. To avoid this confusion, we now make this clear in the updated manuscript.
>
> >**Q**:  I believe the term “supersample” usually refers to the entire set of samples.
>
>  A supersample refers to a single pair $(\rmZ_i^+,\rmZ_i^-)$. The total dataset $\rmZ_{[2n]}$ is composed of n supersamples, i.e., a total of 2n samples. \
> \
> \
> We hope that our response has addressed all your questions and concerns. We kindly ask you to let us know if you have any remaining concerns, and - if we have answered your questions - to reevaluate your score.

---

### Official Review · Reviewer_mpfA · 2023-10-28

**Soundness:** 2 fair
**Presentation:** 3 good
**Contribution:** 3 good
**Rating:** 6
**Confidence:** 4

**Summary:**

This paper studies the class-dependent generalization error through using information-theoretic analysis. The authors begin by demonstrating the necessity of providing class-dependent generalization bounds, highlighting the potential variability in the generalization error across different label classes. They proceed to present class-dependent information-theoretic generalization bounds, including KL divergence-based bound, weight-based CMI bound, $f$-CMI bound and loss-difference CMI bounds. Additionally, empirical results are provided, along with extensions of their findings to other problem settings.

**Strengths:**

Compared with previous works, this paper has the following pros:
1) As demonstrated in its empirical results, previous information-theoretic bounds fail to capture the generalization behavior of individual label classes, while the bounds in this paper overcome such limitations;
2) The class-dependent generalization bounds introduced here can also be used to bound the standard generalization error; 3) The analytical framework of this paper is extendable to other problem settings, such as learning in the presence of sensitive attributes, a domain where previous information-theoretic bounds have not been applied.

In addition, this paper is well-written and easy to follow.

**Weaknesses:**

One main limitation is that the empirical results are confined to ResNet50 on CIFAR10/Noisy CIFAR10. Adding more experimental settings (e.g., SVHN) would further show the effeteness of the proposed bounds.

Furthermore, I have some technical concerns; please see the questions below.

**Questions:**

Major concerns:

1. Regarding Definition 2: Is the class-generalization error defined in Eq.(6) equivalent to Eq.(3) in Definition 1? If so, could you rigorously demonstrate their equivalence?

To my understanding, it is natural to see that Eq.(3) is the same as the following expression:
$$
\mathbb{E}\_{\bf Z}\left[\mathbb{E}\_{U,W|{\bf Z}}\left[\sum\_{i=1}^n\left(\frac{\mathbb{1}\_{\{Y\_i^{-U\_i}=y\}}}{\sum\_{i=1}^n \mathbb{1}\_{\{Y\_i^{-U\_i}=y\}}}\ell(W,Z^{-U\_i}\_i)-\frac{\mathbb{1}\_{\{Y\_i^{U\_i}=y\}}}{\sum\_{i=1}^n\mathbb{1}\_{\{Y\_i^{U\_i}=y\}}}\ell(W,Z^{U\_i}\_i)\right)\right]\right].
$$
By definition, we know that $\sum\_{i=1}^n\mathbb{1}\_{\{Y\_i^{-U\_i}=y\}}+\sum\_{i=1}^n\mathbb{1}\_{\{Y_i^{U_i}=y\}}=n^y_{\bf Z}$, and I think on average we also have $\mathbb{E}\_{U}\left[\sum\_{i=1}^n\mathbb{1}\_{\{Y_i^{-U_i}=y\}}\right]=\mathbb{E}\_{U}\left[\sum_{i=1}^n\mathbb{1}\_{\{Y_i^{U_i}=y\}}\right]=n^y_{\bf Z}/2$. However, this may still not be sufficient to derive Eq. (6). This also raises the question of whether we can use the class-dependent CMI bounds to bound the standard generalization error, i.e. is Corollary 2 a bound for standard generalization error or some other generalization notion?

2. Regarding Corollary 1: Is it tighter than the classic individual mutual information bound in Bu et al. (2019) when $\sigma_Y$ does not depend on $Y$ (e.g., using loss boundedness instead)? I think this might be accurate due to the following:
$$
\mathbb{E}\_{P_Y}\sqrt{D(P_{W,X|Y}||P_{W}\otimes P_{X|Y})}=\mathbb{E}\_{P_Y}\sqrt{\mathbb{E}\_{P_{W,X|Y}}\log\frac{P_{W,X|Y}P_{Y}}{P_{W}\otimes P_{X|Y}P_{Y}}}=\mathbb{E}\_{P_Y}\sqrt{\mathbb{E}\_{P_{W,X|Y}}\log\frac{P_{W,X,Y}}{P_{W}\otimes P_{X,Y}}}\leq \sqrt{I(W;Z)},
$$
where the last inequality is by Jensen's inequality.

If this is correct, you could explicitly state that the individual bound can be recovered from your Corollary 1 or Theorem 1.

3. Could you elaborate more on the last sentence in Remark 2, namely "Therefore, samples from other classes can still affect these bounds, leading to loose bounds on the generalization error of class y"?  As $\max(\mathbb{1}\_{\{Y_i^{-U_i}=y\}}, \mathbb{1}\_{\{Y_i^{U_i}=y\}})=\mathbb{1}\_{\{Y_i^{-U_i}=y \\;{\rm or}\\; Y_i^{U_i}=y\}}$. I understand the other parts in Remark 2. However, I don't understand why samples from other classes contribute to the looseness of the bounds.

4. My previous question also raises another concern regarding the comparison between Theorem 4 and Theorem 3: I agree that $I_{\bf Z}(\Delta_y L_i; U_i)\leq I_{\bf Z}(f_W(X_i^{\pm}); U_i)$; however, I would like to point out that we also have $\max(\mathbb{1}\_{\{Y_i^{-U_i}=y\}}, \mathbb{1}\_{\{Y_i^{U_i}=y\}})I_{\bf Z}(f_W(X_i^{\pm}); U_i) \leq I_{\bf Z}(f_W(X_i^{\pm}); U_i)$ (doesn't this imply that
 $\max(\mathbb{1}\_{\{Y_i^{-U_i}=y\}}, \mathbb{1}\_{\{Y_i^{U_i}=y\}})$ makes Theorem 3 tighter instead of looser?) Therefore, it is uncertain whether Theorem 4 is tighter than Theorem 3 or not. The empirical results suggest that Theorem 4 is tighter, but I hope the authors could clarify why this is expected.

**I would be happy to increase my score if authors could adequately address my main concerns.**

Minor comments:

1. There is a related work that could be included: Hrayr Harutyunyan, et al. "Improving generalization by controlling label-noise information in neural network weights." ICML 2020. In that work, they decompose the mutual information term by chain rule: $I(W;Z)=I(W;X)+I(W;Y|X)$, and using $I(W;Y|X)$ as a regularization term. This might be the first work to incorporate label information into information-theoretic bounds.

2. Notations are not always consistent. For example, in Eq.(6), the selection random variable in the loss function is ${\bf U}_i$ but it becomes ${U}_i$ in the identity function.

3. In Theorem 8 in the Appendix, you may use the widely accepted term for the loss pair-based CMI, namely *evaluated CMI* or *e-CMI*, which was initially introduced in the original CMI paper (see Section 6.2 in the arxiv version of Steinke \& Zakynthinou (2020)).

4. After Eq. (26) and Eq. (48): "Next, Let" ---> "Next, let". There might be more similar typos.

5. When the authors or the publication are not
included in the sentence, the citation
should be in parenthesis using **\citep{}** instead of **\citet{}** or **\cite{}**. Most of citations in this paper are not in parenthesis while the authors or the publication are not part of the sentence. For example, in the first sentence of introduction, He \& Tao
(2020) should be (He \& Tao
(2020)) by using **\citep{}**.

---

> ### Author Response · Authors · 2023-11-22
> **Answers to Reviewer mpfA**
>
> Thank you for your review. In light of our responses to your individual points, please let us know if you have any remaining concerns.
>
> >**Q:** The empirical results are confined to ResNet50 on CIFAR10/Noisy CIFAR10
>
> In addition to CIFAR10//Noisy CIFAR10 results, we also provide results using the MNIST dataset in Appendix C.5, which is a similar digits classification task as SVHN.
>
> >**Q:** Regarding Definition 2: Is the class-generalization error defined in Eq.(6) equivalent to Eq.(3) in Definition 1?
>
> The Reviewer makes a valid point here. Indeed if we use $n^y_{Z_{[2n]}} /2$, taking expectation over Y for eq. (6) does not correspond to the classic standard generalization. To overcome this issue and keep corollary 2, we change the normalization factor $n^y_{Z_{[2n]}} /2$ in Definition 2 to $n P(\rmY=y)$. This does not change the proof of all the results and the standard generalization bound in corollary 2 can be obtained by taking an expectation over $P_Y$ directly. \
> In practice,  $P(\rmY=y)$ is unknown and needs to be estimated using the available data. One simple way is to use $\frac{n^y_{Z_{[2n]}}}{2n}$ as an estimate $P(\rmY=y)$, which is exactly what we did in the empirical sections. Note that our empirical results validate the proposed bounds for this estimated class-wise generalization error. We have made the corresponding changes in the revised manuscript.
>
> >**Q**: Regarding Corollary 1: Is it tighter than the classic individual mutual information bound in Bu et al. (2019)
>
> Good point. Indeed, as shown by the Reviewer, the bound in Corollary 1 is tighter than the bound in Bu et al. (2019) if the sub-Gaussian constant does not depend on Y. This presents an additional contribution to the class-wise analysis. We sincerely appreciate the Reviewer for pointing this out, and we have integrated this result into the revised manuscript. Please refer to Appendix C.4 for further details.
>
> >**Q**: Could you elaborate more on the last sentence in Remark 2..
>
> In the case that one sample in the pair (Z− i , Z+ i ) is from class y and the other is from a different class, the term in the bound is non-zero and the information from both samples of the pair contributes to the bound (I_Z[2n] (fW(X± i ); U_i)). From this perspective, samples from other classes  ($\neq y$) can still affect these bounds, leading to loose bounds on the generalization error of class y.  In the revised manuscript, we now rewrite Remark 2 to clarify this point.
>
> >**Q**: Therefore, it is uncertain whether Theorem 4 is tighter than Theorem 3 or not
>
> Thanks for pointing out this point. Theorem 4 is tighter even if the bound in Theorem 3 has these indicator. This can be seeing that for a fixed Z_[2n], we have 4 different possible cases for each term in the sum of the bounds. We have included the detailed proof as part of  Theorem 4 proof in the revised paper, please refer.
>
> >**Q**: Hrayr Harutyunyan, et al. (2020) is a related work that could be included
>
> Thanks for the reference. We have added it to the related work section of the revised paper.
>
> > **Q**: In Theorem 8, you may use the widely accepted term for the loss pair-based CMI, namely evaluated CMI or e-CMI
>
> Thanks for pointing this out.  We have changed the notation in the updated manuscript and adopted the same notation for the loss-based bound as in  Steinke & Zakynthinou (2020).
>
> >**Q**: Minor comments
>
> Thanks for pointing this out. We have now fixed the mentioned points in the revised manuscript. \
> \
> \
> We hope that our response has addressed all your questions and concerns. We kindly ask you to let us know if you have any remaining concerns, and - if we have answered your questions - to reevaluate your score.

---

> ### Comment · Reviewer_mpfA · 2023-11-23
> **Acknowledgment of the authors' response.**
>
> I would like to thank the authors for their response, and I have briefly read all the reviews and the corresponding responses.  I will read the revised parts more carefully before I discuss with AC and other reviewers (I expected to engage more with the authors before the Review-Author discussion period ends, but there are too many rebuttals submitted today). Below are some quick notes on your response:
>
> Regarding the comparison of the individual bound and Corollary 1, I have no additional questions and would like to share a quick thought that might be helpful: recall the original $I(W;Z)$-based MI bound, by taking a closer look at $W=\mathcal{A}(R,S)$ where random variable $R$ is the randomness of the algorithm $\mathcal{A}$, one can prove a $\mathbb{E}\_{R}\sqrt{I^R(W;Z)}$ based bound, which in some cases can improve over the individual bound, similar bound can be found in [1, Theorem 2], called the "$R$-conditioned bound". In your paper, you focus on $Z=(X,Y)$ in  $I(W;Z)$ and obtain the "$Y$-conditioned bound", serving as a "symmetric" technique to the $R$-conditioned bound (to be clear, unlike $R$ that is independent of $Z$, here $Y$ also dependents on $W$ ). This technique of tightening, extracting an implicit random variable from an explicit one in $I(W;S)$, can be traced back to [2], where the authors extract the subset index from $S$.  Additionally, it would be more interesting if your class-CMI bound (i.e. Theorem 2) can also recover the individual CMI bound but there might be some difficulties arising from Definition 2. This could be a potential theoretical motivation in addition to your practical motivation drawn from the deep learning scenario.
>
> Regarding Definition 2 and the comparision between Theorem 4 and Theorem 3, the revisions look better to me now and I will check more details in Appendix.
>
> [1] Hellström et al. "New family of generalization bounds using samplewise evaluated CMI." NeurIPS 2022.
>
> [2] Negrea et al. "Information-Theoretic Generalization Bounds for SGLD via Data-Dependent Estimates." NeurIPS 2019.

---

### Official Review · Reviewer_Lz6e · 2023-11-03

**Soundness:** 3 good
**Presentation:** 2 fair
**Contribution:** 2 fair
**Rating:** 5
**Confidence:** 4

**Summary:**

The paper addresses focusses on the generalization performance for individual classes rather than the whole data distribution. The authors argue that existing generalization bounds, which typically apply to the average performance across the entire data distribution, do not capture the variations in performance across different classes. To address this gap, the paper introduces a novel information-theoretic bound for class-generalization error using the KL divergence. Additionally, it proposes tighter bounds derived from the conditional mutual information (CMI). The results are supported with experiments on CIFAR dataset.

**Strengths:**

- This is the first work which has introduced class wise generalization bounds (as per the authors) which is an interesting and important idea. They identify that information theoretical based bounds are a natural class of bounds to use for this setting which is interesting.

- The paper is generally well written.

**Weaknesses:**

In my opinion, the main weakness of the paper is the incremental nature of this work. The use of KL divergence and CMI in deriving generalization bounds has been explored in prior literature. The work is essentially following the previous works with this additional conditioning. I don’t see any new technical challenges that appeared due to this conditioning. Neither have the authors mentioned that.

So, the main contribution seems to be showing that information theoretical generalization bounds can be adapted to subsets of data easily.

**Questions:**

- I wanted to ask if there have been any previous works on computing class wise generalization bounds. Or, are there any works on generalization bounds for a subset of dataset with a particular attribute?
- The authors state that information theoretic bounds are natural for this setting as they depend on both the algorithm and the data. It would be useful to discuss this in more detail and discuss why it would be hard to obtain class wise generalization bounds for other types like stability based or hypothesis based. Couldn’t differing property of the subclasses be captured in some way to get different generalization bounds for different classes?

---

> ### Author Response · Authors · 2023-11-22
> **Answers to Reviewer Lz6e**
>
> Thank you for your comments. We provide answers to individual points below.
>
> >**Q:** No new technical challenge
>
>  We respectfully disagree. Please refer to the comprehensive general response provided earlier. As pioneers in the exploration of this tool in such problems, we believe that novelty should not be evaluated solely based on technical complexity.
>
> >**Q:** Previous works on computing class-wise generalization bounds
>
> To the best of our knowledge, this is the first work that considers such a setting, i.e., data with particular attributes, which is one of the key novelties of this paper. Moreover, as shown in Section 4, the developed bounds are useful beyond the scope of class-wise generalization and can provide insights in various contexts.
>
> >**Q:** Information-theoretic bounds are natural for this setting as they depend on both the algorithm and the data
>
> Thanks for pointing this out. It is possible that stability or hypothesis-based bounds can be adopted to capture class-wise generalization. However, we choose information-theoretic bounds because those bounds do not fully characterize all the aspects that could influence the generalization error of a supervised learning problem. For example, VC dimension-based bounds depend only on the hypothesis class, and algorithmic stability-based bounds only exploit the properties of the learning algorithm. As a consequence, both methods fail to capture the fact that generalization error depends strongly on the interplay between the hypothesis class, learning algorithm, and the underlying data-generating distribution. In contrast, information-theoretic bounds incorporate all of these factors into consideration, providing a more holistic understanding of the class-wise generalization problem.
>
> We are happy to answer any further questions you may have. If we have addressed your concerns, we kindly ask that you may consider raising your score.

---

### Author Response · Authors · 2023-11-22
**General answer**

We thank the reviewers for their valuable feedback and time. There seems to be a need for more clarification on the key contributions of this work. Here we would like to highlight the following three main points:

- This paper studies the phenomenon that the generalization errors for the same deep model can differ significantly among different classes by introducing and exploring the concept of “class-generalization error”. To our knowledge, we provide the first rigorous generalization bounds for this concept in the MI and CMI settings. We demonstrate the versatility of our theoretical tools in providing tight bounds for various contexts. In the revised manuscript, we have now reformulated the conclusion to better highlight this point.

- From a technical perspective, while the proof of Theorem 1 can be seen as an extension of prior works by applying the classic Donsker–Varadhan variational representation over a conditional distribution, we note that our proof of Theorems 2-4 requires a combination of Donsker–Varadhan variational along with Hoeffding’s Lemma to obtain tight bounds using the indicator functions in the super-sample setting. To highlight the technical challenges and improve the readability of our proofs, we have restructured the appendix in the revised version by introducing Lemma 3 which encapsulates these key steps and we use it as a core basis for the proofs of Theorems 2-4.
More importantly, we obtain a tighter bound in Theorem 4 without the need for a max operation between two indicator functions. We propose to use the CMI between the label-dependent projection of the loss pair $∆y L$ and the random selection process to upper bound the class generalization error. While this $∆y L$ quantity is inspired by $∆ L$ bound in Wang & Mao (2023), it differs significantly. $∆ L$ simply represents the difference in loss functions and is not well-suited for investigating class-specific generalization. In contrast, $∆y L$ can be interpreted as a weighted sum of class-dependent losses, where indicator functions serve as the weights. This novel projection technique enables us to derive tighter bounds, as demonstrated both empirically and theoretically.

- We also empirically strengthen the findings with supporting experiments validating the efficiency of the proposed bounds. We conducted several empirical experiments with different neural networks in Section 3 and Appendices C.2 C.3 and C.5, where we showed that the proposed bounds can accurately capture the complex behavior of the class-generalization error behavior in different contexts.

In summary, within the revised manuscript, we have highlighted the following modifications:

- To further motivate studying class-wise generalization, we have added a discussion in the introduction on the findings of Balestriero et al. (2022); Kirichenko et al. (2023), where they showed that classical approaches such as data augmentation and regularization, which typically improve standard generalization, are surprisingly harmful for the generalization of certain classes and aggravate this phenomenon.
- Rewrite the conclusion to emphasize the contribution of the paper.
- Restructure the appendix to better present our proofs.
- Add a more elaborated discussion in the related work section and the missing references.
- Fix the typos, inconsistent notation, and referencing style ( \citet vs \citep ).
- Clarify Remark 2.
- Add the results that the bound in Corollary 1 is tighter than the bound in Bu et al. (2019), as suggested by reviewer mpfA.

---

### Meta-Review · Area_Chair_twyc · 2023-12-05

**Metareview:**

This paper derives new generalization bounds for the _class-wise_ risk (i.e., conditioned on the true label). As the paper points out, some models/algorithms do not generalize equally well for all classes. This fine-grained analysis could be useful in cases where one wishes to ensure that the learned model doesn't perform significantly worse on any individual class. The theoretical contributions include an information-theoretic bound (based on KL divergence) and several CMI-based bounds. An empirical study verifies that the bounds capture differences in class-wise risk.

The authors generally agree that the paper is well written, well motivated, and technically sound.

There were some doubts about the assumptions used -- specifically, whether Assumption 1 (symmetric learning algorithm) was too strong, or whether it implied the same bound value for all classes  -- but these were resolved during the reviewer-AC discussions.

A common criticism was that, aside from the class-wise definition of risk, the bounds appear to follow directly from prior work, thereby calling novelty into question. The authors rebutted this in a comment and revised the manuscript such that it highlights the challenges they faced and clarifies how their analysis is novel in overcoming them.

Another potential weakness, identified by `yBBn`, is that there might be a simpler way to derive the bounds using prior work. Unfortunately, since the authors' responses came at the end of the author-reviewer discussion period, `yBBn` was not able to interact with the authors on this point, but provided a proof outline during the reviewer-AC discussions. Because the authors were not able to rebut the claim -- and further, it doesn't seem like a major flaw anyway -- I will not hold this against the paper; just something for the authors to be aware of.

Reviewer `ceHs` had some suggestions to make the plots more informative -- plotting the risk as a function of data size per class -- but I don't see this alone as grounds for rejection; just something to improve in future iterations.

All that being said, there is no getting around the fact that this paper has borderline scores, leaning toward rejection. While the reviews don't highlight any major flaws, none of the reviewers seem all that excited by the results. Perhaps this can be addressed by clarifying the technical novelty and bolstering the discussion section in future iterations.

**Justification For Why Not Higher Score:**

As stated in the meta-review, the reviews do not point to any major flaws in the work, but no one found the results significant enough to give a strong accept.

**Justification For Why Not Lower Score:**

N/A

---

### Decision · Program_Chairs · 2024-01-16

Reject